# Microbiome-derived bile acid signatures in early life and their association with islet autoimmunity

**Santosh Lamichhane** [1,2,3] ✉, **Alex M. Dickens** [1,4], **Tanja Buchacher**[1,3],
**Tianai Lou** [5,6], **Vincent Charron-Lamoureux** [7], **Roosa Kattelus** [1,3],
**Pragya Karmacharya**[1], **Lucas Pinto da Silva**[1], **Matilda Kråkström**[1],
**Omid Rasool** [1,3], **Partho Sen** [1], **Corinn Walker**[8], **Abubaker Patan**[7],
**Emily C. Gentry** [7,9], **Simone Zuffa** [7], **Aron Arzoomand**[10],
**Tadepally Lakshmikanth** [5,6,10], **Jaromir Mikeš** [10], **Aman Mebrahtu**[5,6],
**Tommi Vatanen** [11,12,13,14,15], **Manuela Raffatellu** [8,16,17], **Karsten Zengler**[8,16,18],
**Tuulia Hyötyläinen** [19], **Ramnik J. Xavier** [14], **Petter Brodin** [5,6,10],
**Riitta Lahesmaa** [1,2,3], **Pieter C. Dorrestein** [7,16], **Mikael Knip** [1,11,20] ✉ &
**Matej Orešič** [1,3,21,22] ✉

Emerging studies reveal that gut microbes can conjugate diverse amino acids to bile acids, known as microbially conjugated bile acids. However, their regulation and health effects remain unclear. Here, we analyzed early-life microbially conjugated bile acid patterns and their link to islet autoimmunity. We quantified 110 microbial bile acids in 303 stool samples collected longitudinally (3–36 months) from children who developed one or more islet autoantibodies and controls who remained autoantibody-negative. We identified distinct age-dependent trajectories of these bile acid amidates and correlated them with gut microbiome composition. We found that altered levels of ursodeoxycholic and deoxycholic acid conjugates were linked to islet autoimmunity as well as modulated monocyte activation in response to immunostimulatory lipopolysaccharide and Th17/Treg cell balance. These findings suggest that microbially conjugated bile acids influence immune development and type 1 diabetes risk.

Bile acids (BAs) are cholesterol-derived surfactants produced from cholesterol in the liver, stored in the gall bladder, and released into the small intestine upon meal consumption[1–3]. The primary BAs produced in the hepatocytes consist of cholic acid (CA), chenodeoxycholic acid (CDCA), and their amidates (i.e., either glycine or taurine conjugates)[4]. Gut microbes modify these host-derived primary BAs to a broad range of secondary BAs via a series of reactions, including deconjugation, dehydroxylation, dehydrogenation, oxidation, and epimerization[5]. Almost 95% of these BAs (both primary and secondary) are recirculated to the liver via the enterohepatic circulation across the distal small intestine[6]. The remaining 5% still undergo a wide range of transformations throughout the hind gastrointestinal tract[2,3,7,8]. Traditionally, it was believed that BAs undergo amino acid conjugation only in the liver, mediated by a human enzyme known as bile acid-CoA:amino acid N-acyltransferase[9,10]. However, recent studies demonstrate that in addition to the host, gut microbes can reconjugate BAs via bile salt hydrolase (BSHS) N-acyltransferase activity[11,12]. Therefore, hundreds of microbially conjugated amino acid amidates have been identified and termed as microbially conjugated bile acids (MCBAs)[2,3,7,8].

BAs are known as emulsifiers, facilitating the absorption of lipids in the gut. They are also important signaling molecules and antimicrobial compounds that influence human health. BAs act as ligands to activate various receptors such as farnesoid X receptor (FXR), pregnane X receptor (PXR), Takeda G-protein coupled receptor 5 (TGR5), and sphingosine-1-phosphate receptors (S1PRs)[13–15]. These receptors regulate cellular growth and differentiation, host energy metabolism, and modulate glucose and lipid homeostasis. BAs also play a crucial role in the regulation of the immune system by suppressing proinflammatory cytokine production[16–18]. Dysregulated BA levels are associated with multiple diseases, including type 2 diabetes, inflammatory bowel disease, steatotic liver disease, and various immune-mediated disorders[19].

We previously reported that, at an early age, systemic BAs, including those derived from the secondary BA pathways, are associated with progression to islet autoimmunity and the onset of type 1 diabetes (T1D)[20]. Recently, MCBAs—particularly phenylalanine-, tyrosine- and leucine-conjugated to CA—have been associated with inflammatory bowel disease and cystic fibrosis[7,8]. However, their early-life dynamics and potential role in the development of islet autoimmunity and T1D remain unexplored.

Herein, in a longitudinal cohort study of young children (ages 3–36 months), we investigate the trajectories of MCBAs during early life, including how these microbial BAs are regulated in children who later progressed to islet autoimmunity. The study included children who developed multiple islet autoantibodies (P2Ab) during follow-up and were thus at high risk for progression to T1D later in life[21], the children who developed a single islet autoantibody (P1Ab) but did not progress to T1D during follow-up, and the controls (CTRs), i.e., children who remained islet autoantibody (AAb) negative during the follow-up. We also investigate how the MCBAs modulate the innate and adaptive immune responses, and how they associate with profiles of gut bacterial communities in a prospective series of samples.

## Results

### Presence of MCBAs in early life

We prospectively analyzed MCBAs in a longitudinal series of stool samples ($n = 303$) from 74 children across three study groups: P1Ab, P2Ab, and CTR. For each participant, stool samples were collected at up to six time points for the analysis, corresponding to ages 3, 6, 12, 18, 24, or 36 months (Fig. 1). At the time of sample collection, none of the children had been diagnosed with T1D. We analyzed a total of 110 MCBAs, including BAs (CA, Chenodeoxycholic Acid (CDCA), Deoxycholic Acid (DCA), Ursodeoxycholic Acid (UDCA), and Hyodeoxycholic Acid (HDCA)) conjugated to 22 different amino acids (L-DOPA, Cys, Pro, Met, Asn, Citrulline, Ala, Ser, Arg, Gln, Trp, Val, Tyr, Thr, Lys, His, Phe, Asp, Leu, Ile, Glu, Ornithine), using a targeted LC-MS/MS assay. Mixed BA standards and a pooled quality control (QC) sample to match retention times and MS/MS spectra were analyzed alongside longitudinal stool extracts, to obtain a level 1 annotation.

Among the 110 MCBAs analyzed, 78 were present in at least one of the 303 fecal samples (Fig. 2). Specifically, MCBAs formed from primary BAs, CA, and CDCA, were present in the highest number of fecal samples, while those formed from secondary BAs, i.e., DCA, UDCA, and HDCA, were found in fewer samples. BAs conjugated to Pro, Phe, Glu, His, Ile/Leu, Thr, Trp, Asp, and Tyr were detected in a higher number of samples. In most samples, BA amidates were detected more than once, except for ornithine conjugated to UDCA and L-DOPA conjugated to CA, which were detected in only one fecal sample, respectively. Notably, Ile/Leu isomers were not chromatographically separated. Among all the MCBAs, HDCA conjugates were observed in the lowest number of samples.

### MCBA trajectories in early life

To explore the influences of various factors on early-life MCBA profiles, we conducted multivariate analyses using linear mixed-effect models in samples where breastfeeding information was available ($n = 242$). Age, sex, case status (P1Ab, P2Ab, or CTR), and duration of breastfeeding were treated as fixed effects, with random effects within individual samples/subjects considered. Age emerged as the most significant determinant of MCBA stool profiles compared to sex, case status, and breastfeeding duration. We identified 48 MCBAs that were influenced by age ($p < 0.05$; Fig. 3 and Supplementary Table S1). Additionally, ten MCBAs varied across case groups, nine showed differences between sexes, and three were associated with breastfeeding duration (Supplementary Tables S2–S4). Specifically, primary BA amidates (i.e., CA and CDCA amidates) decreased after the first year of life (Fig. 3a, b). Conversely, MCBAs formed from secondary BAs (DCA and UDCA amidates) exhibited an opposing trend, gradually increasing during the first 18 months of life and remaining stable (DCA-conjugated) or slightly decreasing (UDCA conjugates) at the ages of 24 and 36 months (Fig. 3c, d). Regarding HDCA conjugates, a mixed trend was observed. HDCA-Asp and HDCA-Tyr decreased, while HDCA-Glu increased with age (Supplementary Fig. S1). Here, UDCA-Val, HDCA-Citrulline, CDCA-Phe, CA-His, UDCA-Glu, CA-Ile, CA-Leu, UDCA-Tyr, and UDCA-Phe were affected by sex. Among the 78 detected MCBAs, CDCA-Ala, CA-Pro, and CDCA-Tyr were influenced by breastfeeding status.

### MCBAs in the development of islet autoimmunity

Concentrations of ten MCBAs in stool were found to be different in the children who developed single or multiple autoantibodies during follow-up (P1Ab or P2Ab) when compared to the children in the CTR group who were antibody negative ($p < 0.05$; Fig. 4 and Supplementary Table S2). This included two CA conjugates (CA-Citrulline and CA-Cys), three CDCA conjugates (CDCA-Ala, CDCA-Ser, and CDCA-Tyr), four DCA conjugates (DCA-Ile, DCA-Leu, DCA-Pro, and DCA-Val), and one UDCA conjugate, UDCA-Asn (Fig. 3a–d and Supplementary Table S2). Figure 4 shows the beta coefficients from the linear model. We found that all of the DCA conjugates were lower in the P1Ab group when compared to the CTR, while CDCA-Tyr showed the opposing trend (Fig. 4a, c, d). Likewise, the BA conjugates UDCA-Asn, CDCA-Ser, CA-Citrulline, and CDCA-Ala remained downregulated in the P2Ab group when compared to the CTR group, with the exception of CA-Cys (Fig. 4b–f).

We also compared the MCBA stool profiles separately in different age cohorts. There was no consistent pattern in the same MCBA differences between the study groups (P1Ab, P2Ab, or CTR) when analyzed at individual time points. However, we found that 22 conjugates (12 CA-conjugates, five CDCA-conjugates, four DCA-conjugates, one UDCA-conjugate) remained altered between the P1Ab and CTR groups at either 6, 12, 18, or 36 months of age. Specifically, the CA-conjugates were higher in the P1Ab group (18 months of age), while the DCA-conjugates were generally lower, and other conjugates showed no consistent trend. Additionally, 9 MCBAs were different between the P2Ab and CTR at either 6, 12, or 18 months of age. Notably, UDCA-Asn remained lower in the P2Ab group compared to the CTR group at 12 months of age.

### MCBAs concentrations in the stool are associated with the specific gut microbes

Gut microbes have the ability to conjugate amino acids to BAs. Thus, our next objective was to determine if the microbial conjugated BA profiles, which differed between the study groups (Fig. 4), were associated with bacterial relative abundance in the paired longitudinal metagenomes ($n = 110$) obtained at 3, 6, 12, 18, 24, and 36 months of age from children. Using Spearman correlation analysis, we investigated associations between bacterial composition at the strain level and ten MCBA profiles that were persistently altered in the islet autoantibody group. Our analysis revealed that deoxycholic acid (DCA) conjugates exhibited the most frequent associations with the paired

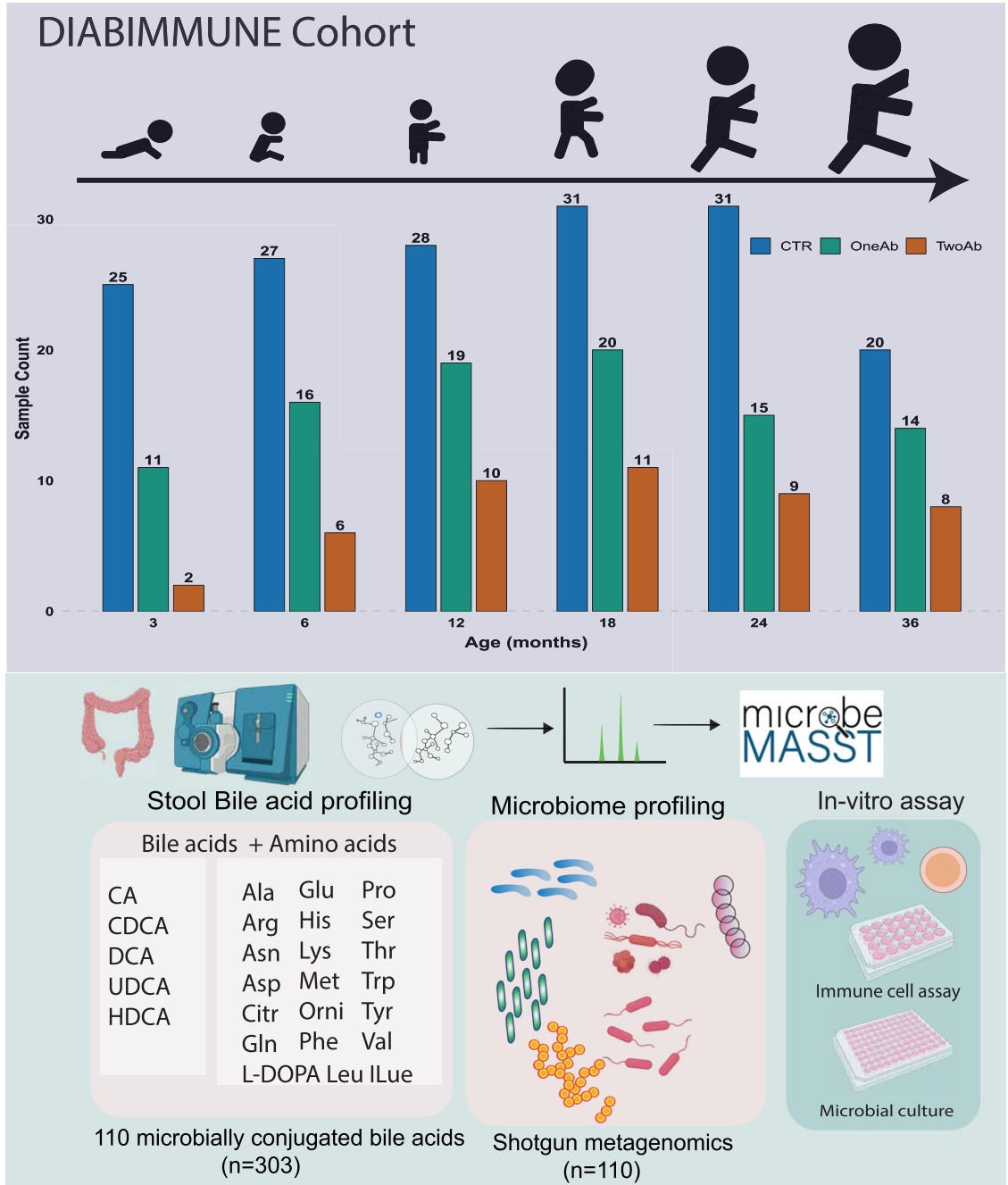

**Fig. 1 | Outline of the study.** We analyzed MCBAs in a longitudinal series of stool samples collected at 3, 6, 12, 18, 24, and 36 months of age from children at HLA-conferred risk for TD, who later developed (i) multiple islet autoantibodies (P2Ab), (ii) single islet autoantibody (P1Ab), or (iii) remained autoantibody negative (CTRs) during the follow-up. The graphical illustration for the in vitro assay, microbes (marked in red), colon, mass spectrometer instrument was created in BioRender. Lamichhane, S. (2025) https://BioRender.com/ crck5np.

microbial strains (Fig. 5). In total, we identified 72 bacteria that were positively or inversely associated with at least one of the DCA conjugates. Notably, *Eubacterium eligens*, *Subdoligranulum unclassified*, *Oscillibacter*, and *Ruminococcus bromii* remained strongly associated with DCA-Val, DCA-Leu, and DCA-Ile, while *Roseburia intestinalis* showed the strongest positive association with DCA-Pro, as well as with *Faecalibacterium prausnitzii*. Furthermore, we observed that many strains possessing BA-transforming capabilities were associated with DCA-Val, DCA-Leu, and DCA-Ile. These strains included *Alistipes onderdonkii*, *Alistipes finegoldii*, *Bacteroides finegoldii*, and *Bacteroides xylanisolvens*. Fifty of the gut bacterial strains were also associated with CDCA conjugates, including *Ruminococcus gnavus*, *Akkermansia*

*muciniphila*, *Escherichia coli*, *F. prausnitzii*, *Erysipelotrichaceae bacterium*, *Bifidobacterium adolescentis*, *Veillonella atypica*, and *Bacteroides fragilis* (Fig. 5 and Supplementary Table S5). Meanwhile, *R. gnavus*, *E. coli*, along with three *Veillonella spp.*, were inversely associated with DCA conjugates (Supplementary Table S5). We also found that UDCA-Asn was associated with *Prevotella oralis*, *Actinobaculum schaalii*, *Haemophilus parahaemolyticus*, and *Morganella morganii*. Among all conjugates, CA-Cys exhibited the strongest positive association with *Eubacterium cylindroides*, *Parabacteroides goldsteinii*, and *Parabacteroides johnsonii*, respectively ($r = 0.57$, $0.47$, and $0.47$).

In addition, we leveraged microbeMASST (https://masst.gnps2.org/microbemasst/) to investigate those MCBAs that were persistently

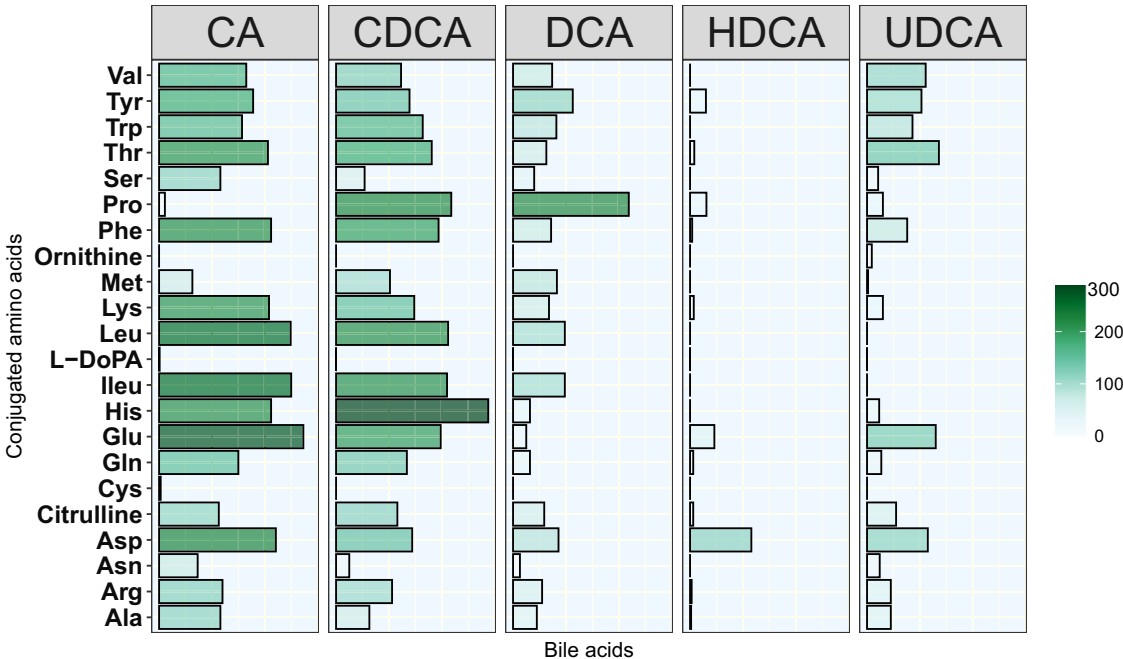

**Fig. 2 | Presence of microbial conjugated bile acids (MCBAs) in fecal DIA-BIMMUNE samples.** The bar in the color map corresponds to the number of fecal samples (*n* = 303) in which MCBAs were detected. CA cholic acid, CDCA cheno-deoxycholic acid, DCA deoxycholic acid, HDCA hyodeoxycholic acid, UDCA urso-deoxycholic acid. The amino acids are Alanine (Ala), Arginine (Arg), Asparagine (Asn), Aspartic Acid (Asp), Cysteine (Cys), Glutamic Acid (Glu), Glycine (Gly), His-tidine (His), Isoleucine (Ile), Leucine (Leu), Lysine (Lys), Methionine (Met), Phenyl-lalanine (Phe), Proline (Pro), Serine (Ser), Threonine (Thr), Tryptophan (Trp), Tyrosine (Tyr), and Valine (Val). L-DOPA L-3,4-dihydroxyphenylalanine.

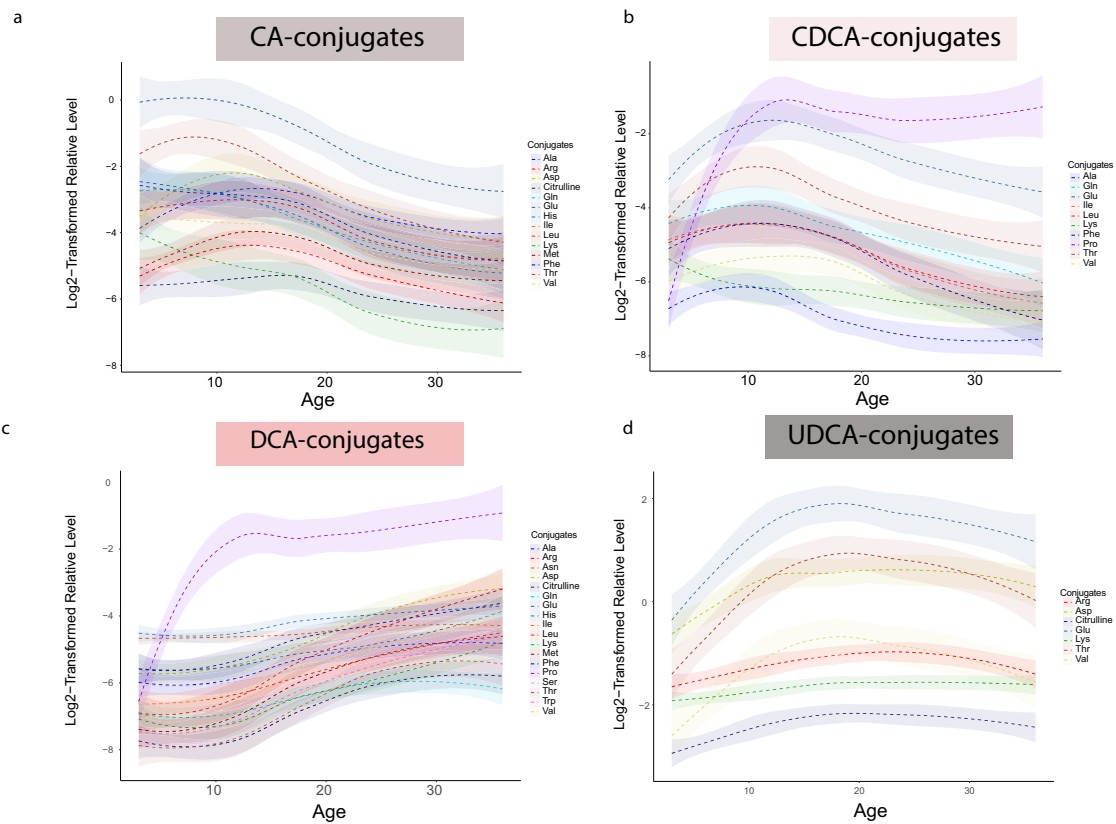

**Fig. 3 | Trajectories of MCBA in early life.** The loess curve plot of MCBAs over time (*n* = 303, fecal samples over 3, 6, 12, 18, 24, and 36 months) for **a** CA-conjugates, **b** CDCA-conjugates, **c** DCA-conjugates, and **d** UDCA-conjugates. The shaded area around each smooth line 3a–d is the 95% confidence interval for the smoothed curve. CA cholic acid, CDCA chenodeoxycholic acid, DCA deoxycholic acid, HDCA hyodeoxycholic acid, UDCA ursodeoxycholic acid.

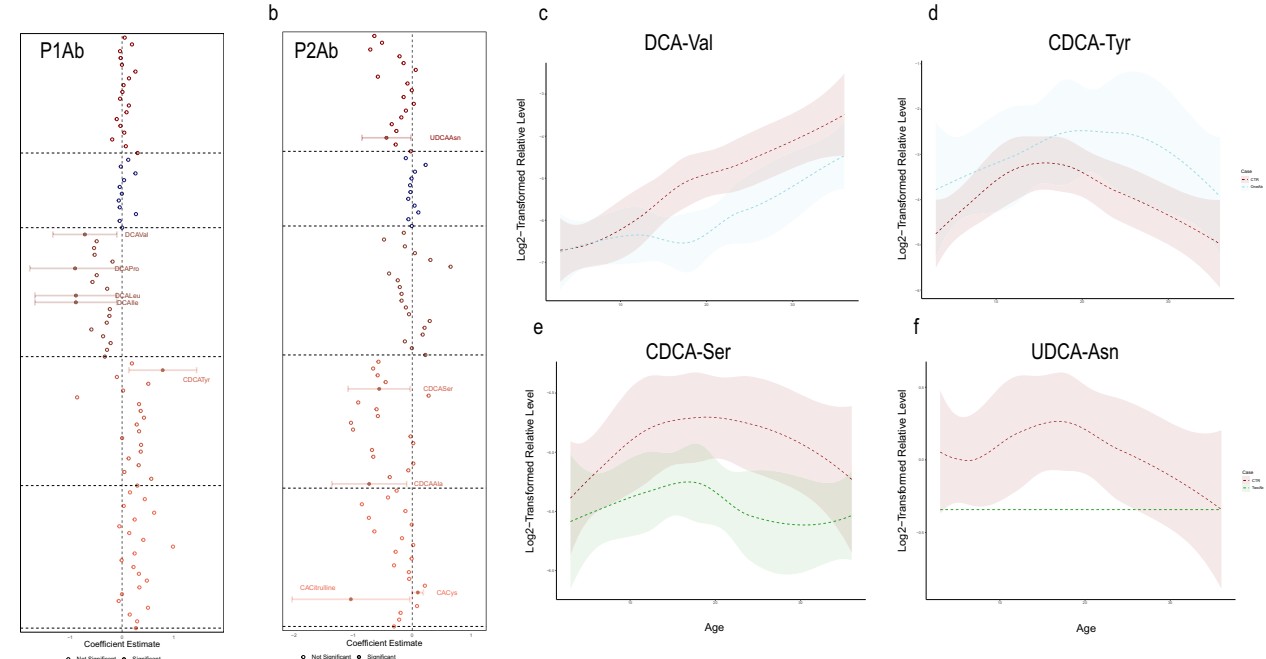

**Fig. 4 | MCBAs in progression to islet autoimmunity.** Forest plot illustrating the coefficient estimate of a linear mixed-effects model for individual MCBAs species, with fixed covariates of case **a** CTR ($n = 131$) vs **b** P1Ab ($n = 73$), CTR ($n = 131$) vs P2AB ($n = 38$), age, sex, and length of breastfeeding accounting for random effects within individual samples ($n = 242$). Filled circles with corresponding confidence intervals (error bars) represent significant MCBA species ($p < 0.05$), while faded circles depict non-significant MCBAs. The $p$-values shown are nominal; adjusted $p$-values (corrected for multiple comparisons using the Benjamini–Hochberg method) are available in Supplementary Table S2. **c–f** The loess curve plot of MCBAs over time for significant MCBA species obtained in the linear mixed-effects model ($p < 0.05$). The shaded area around each smooth line 3a–d is the 95% confidence interval for the smoothed curve. CA cholic acid, CDCA chenodeoxycholic acid, DCA deoxycholic acid, HDCA hyodeoxycholic acid, UDCA ursodeoxycholic acid. The amino acids are Alanine (Ala), Asparagine (Asn), Cysteine (Cys), Isoleucine (Ile), Leucine (Leu), Lysine (Lys), Proline (Pro), Serine (Ser), Tyrosine (Tyr), and Valine (Val).

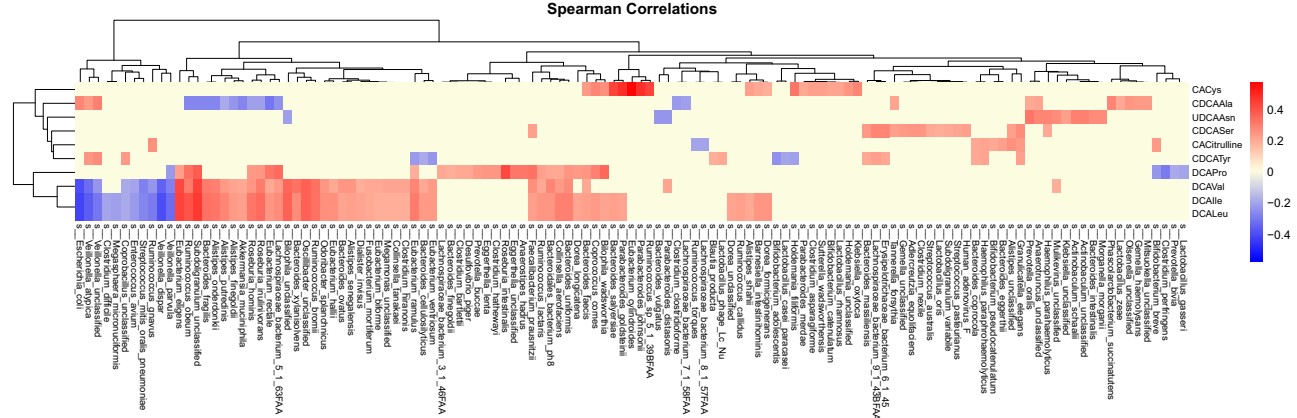

**Fig. 5 | Cross-correlation between the microbes and stool levels of selected MCBAs.** Heatmap showing the correlation coefficients of association between microbes and stool levels of MCBAs that were found altered in progression to islet autoimmunity, in a subset of the sample with available metagenomics data ($n = 110$). Red color represents positive correlations, while blue represents negative correlations, as determined by Spearman's rank correlation. Only correlation coefficients with nominal $p$-values less than 0.05 are shown. The adjusted $p$-values are shown in Supplementary Table S7.

altered in the islet autoantibody group and to identify possible microbial producers[22]. We found that CDCA-Ala was predominantly found in cultures of *Clostridium scindens* ATCC 35704 and *Clostridium sordelli AO32*, UDCA-Asn in *R. gnavus* ATCC 29149. CA-Citrulline was found in a diverse number of microbial cultures, including *Enterococcus faecium* 513, *Bifidobacterium breve* HPH0326, and 21 other microbial strains (Supplementary Table S6). Likewise, DCA-Ile/Leu was detected in *E. coli*, *B. adolescentis* L2-32, *Bifidobacterium angulatum* F16_22, and several other strains (Supplementary Table S6). Likewise,

microbial candidates for CDCA-Ser and DCA-Val are listed in Supplementary Table S6.

Based on multi-omics analysis and strain availability, we cultured nine predicted BA metabolizers that were altered in the P2Ab group (*A. onderdonkii*, *R. bromii*, *Clostridium bartlettii*, *Bacteroides vulgatus*, *Bacteroides wadsworthia*, *Coprococcus comes*, *Parabacteroides distasonis*, *Bacteroides intestinalis*, and *Eggerthella lenta*) in brain heart infusion media supplemented with conjugated UDCA-Asn, UDCA, and Asn, as well as in control media without supplements

(Supplementary Fig. S2). After 72 h of culturing, samples were analyzed using LC-MS/MS and compared to the UDCA-Asn reference. Monocultures of *C. bartlettii*, *B. vulgatus*, and *C. comes* had UDCA-Asn deconjugation potential. *C. bartlettii*, *C. comes*, and *R. bromii* had conjugation potential. In addition, *A. onderdonkii* and *E. lenta* supplemented with UDCA-Asn, we observed an increase in the UDCA-Asn signal (Supplementary Fig. S2). These findings confirm that specific human gut microorganisms are capable of producing these conjugated MCBAs.

## MCBAs modulate host immune responses

We further explored the potential of MCBAs to modulate the innate immune system, specifically examining whether the conjugated form of available UDCA and the unconjugated BAs UDCA and DCA exert immunomodulatory properties in vitro in response to lipopolysaccharide (LPS) using human whole blood cultures. Whole blood from healthy donors was diluted 1:1 in WB-STIM buffer (Cytodelics AB, Stockholm) and stimulated with a Salmonella-derived form of immunostimulatory LPS. We then measured three signaling pathways downstream of the TLR4 receptor in a time-course detecting phosphorylation of p38 (pp38) and ERK (pERK), as well as degradation of IkBa, at the single-cell level by intracellular staining and flow cytometry (Fig. 6a). To robustly compare immunomodulatory effects of different BA conjugates across the entire time course we calculated the area under the curve (AUC) for each signaling intermediate and compared theses across pretreatment conditions and used combinations of small molecule inhibitors of ERK and p38 phosphorylation as negative controls. The different UDCA conjugates exerted variable effects (Fig. 6b). Unconjugated UDCA, Asn-UDCA, and Cit-UDCA partially inhibited LPS-induced signaling across all three pathways investigated, while Trp-UDCA, Asp-UDCA, and Glu-UDCA enhanced the LPS-induced signaling (Fig. 6b), indicating immunomodulatory potentials of these discovered MCBAs. DCA strongly inhibited LPS-induced signaling across all three signaling pathways in human blood monocytes and was even more efficient than either of the conjugated UDCAs tested (Fig. 6c). Given the abundance of LPS in the human intestine and the previously reported role of LPS exposure early in life in relation to T1D development[23] it is intriguing that BA modulation of LPS-induced responses offers an additional, previously unrecognized layer of regulation for the establishment of healthy immune-microbe interactions early in life.

Next, we sought to investigate whether MCBAs interact with the adaptive immune system. Specifically, we examined the effects of MCBAs during the early in vitro differentiation stages of human Th17 and in vitro induced Treg (iTreg) cells. Naïve CD4 + CD25- T cells were isolated from human cord blood and cultured with MCBAs under conditions promoting Th17 and Treg differentiation (Fig. 7a). We screened three MCBAs—UDCA-Asn, CDCA-Tyr, and CDCA-Ser, as well as unconjugated UDCA, which we identified as altered in the P2Ab and P1Ab groups compared to the CTR (Fig. 4). Our results showed that UDCA-Asn and CDCA-Ser enhanced Th17 cell differentiation, leading to increased IL-17a secretion (Fig. 7b, c). Conversely, these compounds inhibited the differentiation of iTreg cells, as evidenced by a decrease in Foxp3 expression levels (Fig. 7f–g). However, while neither unconjugated UDCA nor CDCA-Tyr affected the iTreg population (Fig. 7h, i), both compounds reduced IL-17a secretion, thereby impeding Th17 differentiation (Fig. 7d, e). Further, we assessed RORC, as well as RORA expression in T cells treated with Asn-UDCA, Ser-UDCA, Tyr-CDCA, and UDCA. Notably, we observed that RORA expression positively correlated with IL-17A levels in Th17 cells treated with these MCBAs (Supplementary Fig. S4), suggesting a potential role for RORα in bile acid-mediated modulation of Th17 cells. However, RORC expression or correlation with IL-17A secretion remained unchanged.

## Discussion

By comprehensive analysis of fecal MCBAs in a longitudinal birth cohort, we demonstrated that MCBAs display a specific age-dependent profile during the first three years of life. Certain BA conjugates, specifically primary BA amidates (CA and CDCA), decreased over time, while the secondary BA amidates (DCA and UDCA) increased during early life, before stabilizing. Previous metabolomics studies suggest that age has a significant impact on the longitudinal trajectories of metabolites. Several studies previously found that systemic lipids, including BAs, show distinct age-related trajectories[20,24]. However, to our knowledge, this is the first study to define the trajectories of MCBAs in the human gut during infancy. Bacteria from the human gut produce BSHs that re-amidate BAs to generate MCBAs[11,12]. During the first three years of life, gut microbial maturation is a dynamic process, that can consequently shape the MCBA profiles and thus explain our observations.

The frequencies of MCBA conjugations varied depending on the specific BA type. Notably, the amidates of HDCA and UDCA were the least frequently observed amino acid conjugations. The composition of BAs in humans is markedly influenced by the gut microbiome[2,25]. Additionally, the human gut microbiome composition differs both within and between individuals[26,27], which can influence the amidate conjugations. Recent in vitro studies that investigated bacterial species commonly found in the human intestinal tract, have revealed varying abilities among the gut microbial strains to perform these amidate conjugations[5], thus suggesting a complex interplay between microbial diversity and MCBA metabolism within the gastrointestinal ecosystem. Our data also suggest that gut bacteria contribute to the production of these MCBAs.

We found that children with human leukocyte antigen (HLA)− conferred risk for T1D, who later progressed to single or multiple islet autoantibodies in the follow-up period, have a distinct MBCA profile compared to those who remained autoantibody negative. Previously, we demonstrated that at-risk children exhibit persistently altered levels of both host-derived systemic BAs (including both glycine and taurine conjugates) and host-microbial BA co-metabolism, compared to children who develop at least one single islet autoantibody or remain negative for islet antibodies during follow-up[20].

BAs are recognized as immunoregulatory metabolites[16] and microbially-derived secondary BAs are crucial for the maintenance of immune system homeostasis[28]. Hang et al. identified that microbially-derived secondary BAs, including 3-oxoLCA and isoalloLCA, affect host immune responses by directly modulating T-cell differentiation[16]. Moreover, the levels of isoallo LCA were reported to be altered in patients with IBD[28]. Gentry et al. discovered that some of these newly discovered MCBAs, particularly CA conjugated to Glu, Ile/Leu, Phe, Thr, Trp, or Tyr, could be associated with IBD. In the latter study, for CDCA-Met, DCA-Met, CDCA-Phe, CDCA-Trp, and CDCA-Tyr, a notable increase in the levels of interferon-γ (IFNγ) was reported, with CDCA-Trp showing a sixfold rise[8]. IFNγ is a cytokine with essential roles in regulating immune homeostasis and inflammatory responses in humans. In line with others, our data suggest that MCBAs modulate immune responses by influencing the differentiation process of Th17 and Treg cells.

It is well established that several secondary bile acids act as inverse agonizts of RORγt[16,18], a transcription factor essential for Th17 lineage commitment. In this study, we assessed the expression of RORC and RORA in T cells treated with Asn-UDCA, Ser-UDCA, Tyr-CDCA, and UDCA. Notably, RORA expression positively correlated with IL-17A levels in Th17 cells exposed to the tested bile acids, suggesting a potential role for RORα in bile acid-mediated Th17 modulation. While RORα has previously been studied in the context of circadian regulation and bile acid synthesis[29], it is also a ligand-responsive nuclear receptor that may be directly influenced by bile

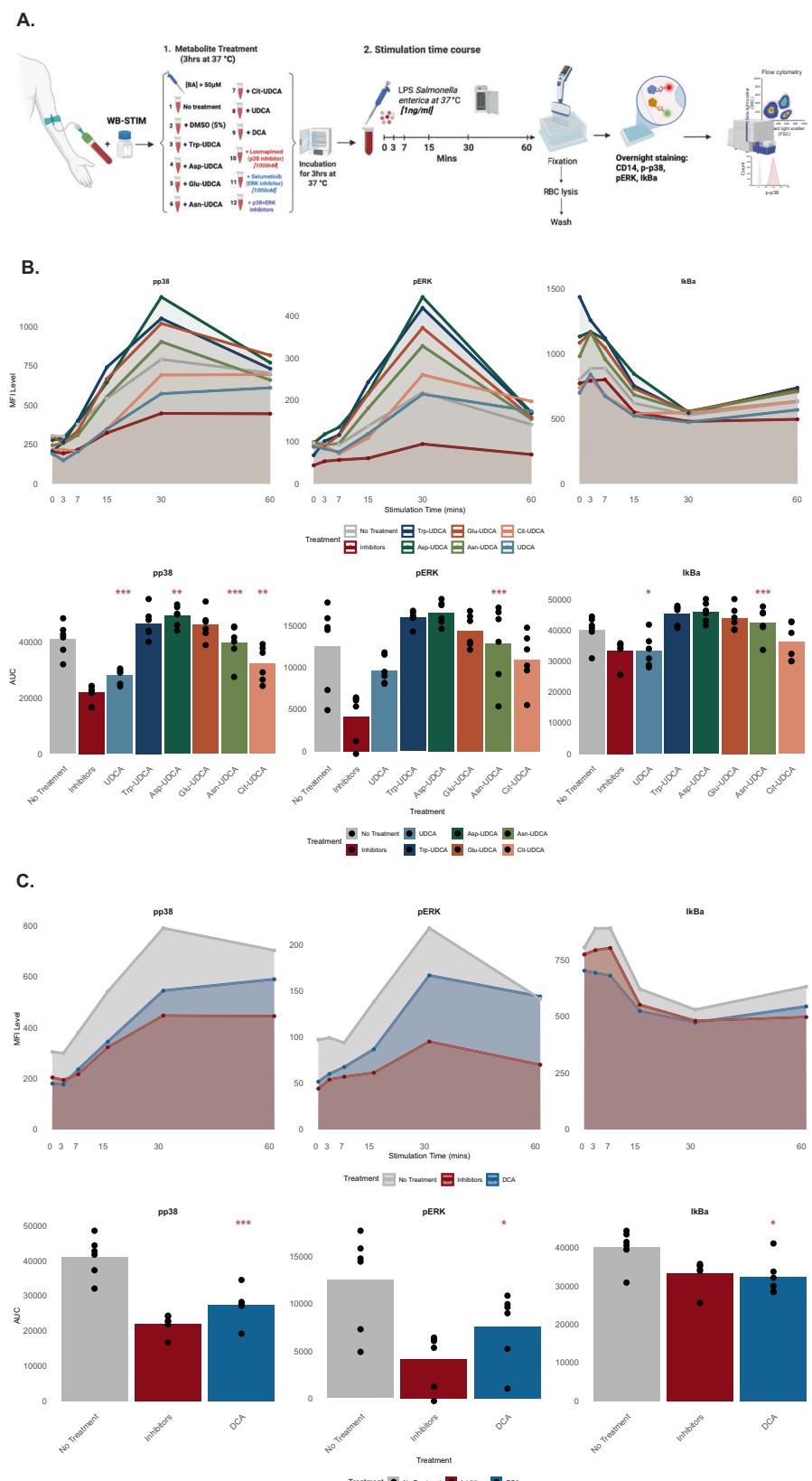

acid derivatives, independent of circadian cues. We hypothesize that MCBAs regulate RORα either directly or through upstream metabolic signaling pathways, thereby promoting IL-17A expression. Given the critical roles of Th17 and Treg cells in inflammatory diseases and their close association with gut bacteria[30], our study suggests molecular pathways linked to emerging microbiota-derived bioactive

compounds that modulate T-cell function. Together, our findings point to a potential RORα–IL-17A axis in MCBAs-driven immune regulation and underscore the need for future studies to elucidate the upstream mechanisms modulating the RORα activation.

The sensing of LPS and the degree to which different LPS variants stimulate immune cell responses early in life has previously been

**Fig. 6 | Effects of secondary bile acids on LPS-induced signaling in human monocytes. a** Time-course analysis of LPS-stimulated pp38 and pERK and degradation of IkBa, markers of canonical NF-kB activation, in primary human monocytes (*n* = 6). LPS (1 ng/mL) induced robust activation of pp38/pERK and IkBa degradation within 60 min. Geometric MFI was measured by flow cytometry, and the average of *n* = 6 was plotted in time curves. Quantitative differences among treatments were shown by AUC analysis. Geometric MFI values were used for AUC calculations. "Inhibitors" serve as a positive control for pathway inhibition, and "No treatment" represents the baseline LPS response. **b** UDCA and its amino-acid conjugates (Trp-UDCA, Asp-UDCA, Glu-UDCA, Asn-UDCA, Cit-UDCA) show limited or no inhibition of these pathways. **c** DCA consistently inhibits signaling across all three pathways. DCA demonstrates significant suppression of LPS-induced

signaling. AUC values were analyzed by one-way ANOVA followed by post-hoc pairwise comparisons vs no treatment using the general linear hypothesis test (*glht*, R package *multcomp*). All tests were two-sided, with *p*-values adjusted for multiple comparisons using the Benjamini–Hochberg false-discovery-rate (FDR) method. Significant effects included inhibition of pp38 by Cit-UDCA (*p* = 0.0065), UDCA (*p* = 0.00011), and DCA (*p* = $7.37 \times 10^{-5}$) and activation by Asn-UDCA (*p* = $7.20 \times 10^{-10}$); activation of pERK by Asn-UDCA (*p* = 0.00031) and inhibition by DCA (*p* = 0.0378); elevation of IkBα by Asn-UDCA (*p* = $3.11 \times 10^{-13}$) and reduction by UDCA (*p* = 0.0309) and DCA (*p* = 0.0185). Significance: ***\*\*\*p* < 0.001; *\*\*p* < 0.01; *\*p* < 0.05. Illustrations were Created in BioRender. Tadepally, L. (2025) https://BioRender.com/5e1kt0e. DCA deoxycholic acid, UDCA ursodeoxycholic acid. The amino acids are Citrulline (Cit), Asparagine (Asn), and Glutamic Acid (Glu).

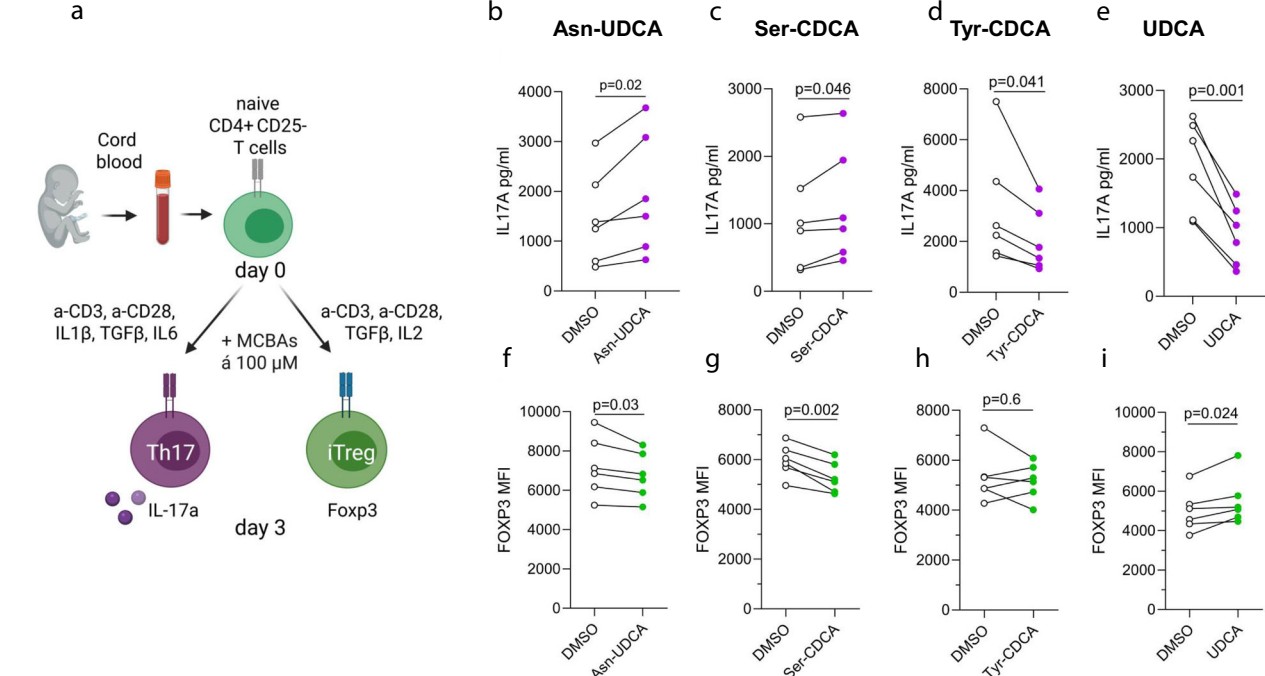

**Fig. 7 | MCBAs modulate Th17 and Treg cell differentiation. a** Schematic of the Th17 and iTreg differentiation protocol for primary human naïve CD4⁺CD25⁻ T cells isolated from the umbilical cord blood of healthy neonates. CD4⁺CD25⁻ T cells were activated with anti-CD3/anti-CD28 and differentiated into Th17 or iTreg cells in the presence of corresponding cytokines for 3 days. DMSO control or conjugated bile acids (Asn-UDCA, Ser-CDCA, Tyr-CDCA, and unconjugated UDCA) at 100 μM were added on day 0 of differentiation. Illustrations in a panel were created with BioRender.com (licensed). **b–e** IL-17a secretion in the supernatant of Th17 cultures treated with Asn-UDCA (**b**), Ser-CDCA (**c**), Tyr-CDCA (**d**), and UDCA (**e**) was

quantified on day 3 of differentiation from four biological replicates using ELISA. **f–i.** Intracellular Foxp3 protein expression in iTregs cultured with Asn-UDCA (**f**), Ser-CDCA (**g**), Tyr-CDCA (**h**), or UDCA (**i**) was assessed on day 3 of differentiation by flow cytometry. Geometric mean fluorescence intensity (MFI) values are shown for six biological replicates. Statistical significance was determined using a paired, two-tailed Student's t-test. Illustrations were Created in BioRender. Hirvonen, K. (2025) https://BioRender.com/5s3bnh7. Asn–UDCA asparagine–ursodeoxycholic acid, Ser–CDCA serine–chenodeoxycholic acid, Tyr–CDCA tyrosine–chenodeoxycholic acid, UDCA ursodeoxycholic acid.

associated with T1D in this cohort[23]. Here, we show that secondary BAs variably modulate LPS-induced signaling, thus providing an additional layer of regulation conferred by microbiota members. We demonstrate that conjugated forms of UDCA display varying effects on the activation of monocytes, following exposure to immunostimulatory LPS derived from Salmonella in vitro. These results indicate that the presence of immunostimulatory vs immunoregulatory LPS variants in variable proportions, together with the abundance of immunoregulatory BA conjugates, determines the overall potential of LPS to trigger inflammatory responses. This complements the actions of other immunoregulatory metabolites such as indole-3-lactic acid and other tryptophan metabolites, which are also known to modulate inflammatory responses and T cell development in human newborns[31]. Thus, we suggest that disturbances in the gut microbiome-BA axis during early life alter immunomodulation and potentially contribute to the initiation and/or progression of islet autoimmunity.

We acknowledge some limitations in our study. The first is the small sample size, which limited our ability to systematically assess the impact of MCBA through multiple comparisons. However, to the best of our knowledge, this is the first study to report the dynamics of MCBA during early life in humans and its association with islet cell autoantibodies. Our results offer potentially clinically significant insights into the microbe-host interplay involved in immune dysregulation; however, the causal relationship remains unclear and warrants further investigation. Additionally, due to limited sample availability, we were unable to measure circulating levels of MCBA during early life. Understanding the systemic bioavailability of these metabolites in the blood is essential to elucidate their role in peripheral immune modulation. Despite this limitation, we conducted in vitro immune assays to explore how these bile acids influence host immune responses. Specifically, we found that altered levels of ursodeoxycholic and deoxycholic acid conjugates modulated monocyte activa-

tion in response to immunostimulatory LPS and affected the Th17/Treg cell balance. Our data also suggest that the tested MCBAs modulate the RORA expression in T cells; however, the underlying mechanism of how the tested bile acids influence intracellular signaling cascades remains unexplored. Taken together, our study presents the first known exploration of MCBAs' dynamics in early life, revealing their potential role in shaping islet autoimmunity. Our findings show that MCBA levels in stool are closely linked with the gut microbiome and influence the differentiation of Th17 and Treg immune cells, highlighting MCBAs as important modulators of immune development.

## Methods

### Study subjects

The DIABIMMUNE study recruited 832 families in Finland (Espoo), Estonia (Tartu), and Russia (Petrozavodsk) with infants carrying HLA alleles that confer risk for islet autoimmunity and T1D. The subjects involved in the current study were chosen from the subset ($n = 74$) of available samples (stool) in the international DIABIMMUNE study. This comprises children who progressed to at least a single AAb (P1Ab, $n = 24$) or to multiple islet AAb (P2Ab, $n = 12$), and controls (CTRs, $n = 38$), i.e., the children who remained islet AAb-negative during the follow-up. Samples were collected longitudinally at 3, 6, 12, 18, 24, and 36 months from each child. Here, no prior sample-size estimation was performed. The study groups were matched for HLA-associated diabetes risk, sex, country, and period of birth.

This study was conducted according to the guidelines in the Declaration of Helsinki. The Ethics and Research Committee of the participating Universities and Hospitals approved the study protocol. All families provided written informed consent prior to sample collection.

### Quantification of bile acids

The BAs were measured in fecal samples as described previously[20]. All fecal samples were freeze-dried prior to extraction to account for the inconsistency in the fecal water content and dry weight in the stool. The fecal homogenate was prepared using fecal samples and ethanol in 2:1 $w/v$ ratio. The MCBAs were extracted by adding a volume of 10 μL of fecal homogenate to 200 μL of crash solution containing MeOH and 25ppb of internal standard mix: CA-d4, CDCA-d4, UDCA-d4, DCAd4, LCA-d4, TCA-d4, GUDCA-d4, GCDCA-d4, GDCA-d4, GCA-d4, TLCA-d4, TDCA-d4, TCDCA-d4, TUDCA-d4, TCA-d4, and GLCA-d4. The samples were filtered using a 96-well protein precipitation filter plate (Supelco). The filtrate was transferred to a glass vial and dried with N2 flow at 45 °C and resuspended in 50 μL of resuspension solution (methanol:water 4:6). For QC samples, a pooled sample was prepared in the same way.

The analyses were performed on a Sciex 6600 quadrupole time-of-flight mass spectrometer (Sciex, United States) coupled to a Sciex Exion LC system. Separation was performed on an ACQUITY PREMIERE HSS T3 (2.1 × 100 mm, 1.8 μm) column, Waters, using gradient elution. The eluents were water/methanol (7:3 $v/v$) with 2 mM ammonium acetate (A) and methanol with 2 mM ammonium acetate (B), the flow rate was 0.4 mL/min, the injection volume was 5 μL, and the column was kept at 45 °C. The gradient program was 0–0.5 min: 5% B, 4 min: 60% B; 11–16 min: 100% B, 16.1–17 min 5% B. The mass spectrometer was operated in negative mode. Full scan in the range 100-800 m/z was used for all samples, except QC samples, where information-dependent analysis was performed. The source parameters CUR, GS1, GS2, ISFV, and TEM were 25, 40, 40, 4500, and 650, respectively. The accumulation time was 250 ms. The mass was calibrated using sodium formate clusters between every five samples.

Identification of MCBAs was carried out by comparing retention time and accurate mass (mass error < 10 ppm) to authentic standards, which were synthesized as described previously[8]. Further, for the QC samples, the fragmentation pattern was compared with the authentic standards. Data was processed using SciexOS (3.0). Results were reported as the peak area of the MCBAs normalized with the corresponding internal standard. For QC, we randomized the order of samples and injected (1) a pooled QC, (2) a blank sample, and (3) a known standard every ten samples. In addition to that, the samples were blinded to the person preparing and running the experiments.

### Analysis of islet autoantibodies

Four diabetes-associated autoantibodies were analyzed from each serum sample with specific radiobinding assays: insulin autoantibodies (IAA), glutamic acid decarboxylase antibodies (GADA), islet antigen-2 antibodies (IA-2A), and zinc transporter 8 antibodies (ZnT8A) as described previously in ref. 20. The cut-off values for autoantibody positivity were based on the 99th percentile in non-diabetic children and were 2.80 relative units (RUs) for IAA, 5.36 RU for GADA, 0.78 RU for IA-2A, and 0.61 RU for ZnT8A.

### Gut microbiome analysis by shotgun metagenomics sequencing

Metagenomic shotgun sequencing and data processing were conducted as previously described in refs. 20,23,32–34. Raw metagenomic sequencing data were retrieved from (https://diabimmune.broadinstitute.org/) (NCBI BioProject ID: PRJNA231909). Stool samples ($n = 110$) were common between the published metagenomics data and the stool BAs measured in the present study. Metagenomic data from the matched samples ($n = 110$) were considered for further analysis. As stated earlier[20], host genome–contaminated reads and low-quality reads are already removed from the raw sequencing data using kneadData v0.4. Taxonomic microbiome profiles were determined using MetaPhlAn2 using default parameters as described[20].

### Bacterial cultures screening

All bacterial cultures were started from glycerol stock and incubated at 37 °C for three days in an anaerobic chamber (10% $CO_2$, 7.5% $H_2$, 82.5% $N_2$) in a filtered BHI medium at a pH adjusted to 7.2 using 5 N NaOH. After the cultures were normalized at $OD_{600} = 0.02$, 200 μL of bacterial suspension was added in triplicate in a 96-well plate with and without 100 μM of Asn-UDCA and incubated for 72 h at 37 °C. Bacterial cultures were also extracted at the start of the experiment (0 h) to establish a baseline. Following bacterial growth, 200 μL of culture was transferred to a new 2 mL deep-well plate, 600 μL of 50% MeOH/$H_2O$ was added, and incubated overnight at 4 °C. Samples were centrifuged at 2000 RPM for 10 min, and 200 μL was transferred into a deep-well plate and dried overnight in a CentriVap and stored at −80 °C until LC-MS/MS analysis.

### LC-MS/MS analysis of bacterial culture

The samples were resuspended in 200 μL of 50% MeOH/$H_2O$ with sulfadimethoxine as internal standard and incubated overnight at −20 °C. Samples were centrifuged at 2000 RPM for 10 min, and 150 μL was transferred to a shallow 96-well plate. A pooled sample was created for QC. Samples were randomized and analyzed using a Vanquish UHPLC (ultra-high performance liquid chromatography) system coupled to an Orbitrap Exploris 240 mass spectrometer (Thermo Fisher Scientific). The chromatographic separation was achieved by using a Phenomenex polar C18 column (2.6 μm particle size, 100 × 2.1 mm), and the mobile phase consisted of $H_2O$ + 0.1% FA (solvent A) and ACN + 0.1% FA (solvent B). Five microliters of samples were injected and eluted using the following gradient: 0–0.5 min 5% B, 0.5–1.1 min 5% B, 1.1–7.5 min 40% B, 7.5–8.5 min 99% B, 8.5–9.5 min 99% B, 9.5–10 min 5% B, 10–10.5 min 5% B, 10.5–10.75 99%B, 10.75–11.25 99% B, 11.25–12 min 5% B. Data-dependent acquisition (DDA) mode was used to acquire the MS/MS data using positive electrospray ionization (ESI+). Sheath gas was set to 50 L/min, aux gas flow rate was set at 10 L/min, and sweep gas was set to 1. The spray voltage was 3.5 kV, ion transfer tube 325 °C, and vaporizer temperature 350 °C. The AcquireX Deep Scan method was

**Table 1 | Fluorescent marker antibodies (surface and intracellular) used in flow cytometry**

| Fluorophore | Marker | Catalog number | Antibody dilution, times | Clone | Vendor |
|---|---|---|---|---|---|
| BV650 | CD14 | 301835 | 5000 | M5E2 | Biolegend |
| PE-Cy7 | p-p38 | 25-9078-42 | 5000 | 4NIT4KK | Invitrogen |
| PE-DAZZLE | p-ERK | 369518 | 5000 | 6B8B69 | Biolegend |
| 488 | IκBα | 5743S | 200 | L35A5 | Cell Signaling Technologies |

enabled, and an exclusion list was created by injecting four times the pooled sample. MS/MS scan range was set to 100–1000 $m/z$, RF lens (%) was 70, a resolution of 60,000 with 1 microscans, a charge state of 1, the expected peak width was 6 s, and advanced peak determination was enabled. The automatic gain control (AGC) target was set to standard, and with a maximum injection time set to auto. Dynamic exclusion was set to custom, and the following parameters were used: Exclude after number of times: 2, if it occurs within 3 s, with a duration of 4 s. Isotopes were excluded. Up to 10 scans per MS1 were collected with a resolution at 200 $m/z$ of 22,500 with 1 microscans. The isolation window was set to 1 $m/z$. The AGC target was set to custom with 200% the normalized AGC target, with a maximum injection time set to Auto. The scan range mode was set to Auto. The collision energies were set to a stepwise increase of 25, 40, and 60 eV.

## LC-MS/MS data processing

Thermo RAW files were converted to.mzML using the ProteoWizard MSconvert software. Feature detection and extraction were performed using MZmine 4.4.3[35]. The batch file used for feature extraction (.mzbatch) can be found on the GitHub webpage. The data was imported using the MS1 and MS2 detectors using the factor of the lowest signal of 3 and 2, respectively. Mass detection was also performed using the above parameters. For the chromatogram builder, the minimum consecutive scans was set to 4, intensity to 3E4, height to 1.5E5, and 10 ppm for $m/z$ tolerance. Smoothing was applied using the Savitzky–Golay algorithm before applying the local minimum feature resolver, which had the following parameters: chromatographic threshold set to 90%, minimum search range retention time of 0.05 min, minimum ratio of peak top/edge of 2, and a minimum of 4 scans. Then, the $^{13}C$ isotope filter and isotope finder were applied using a $m/z$ tolerance of 3 ppm and retention time tolerance of 0.04 min. Features were aligned using the join aligner module at a $m/z$ tolerance of 10 ppm and retention time tolerance set to 0.07 min. The feature list rows filter was applied using two samples or 10%. Peak finder was set to an intensity tolerance of 20%, a $m/z$ tolerance of 10 ppm, retention time tolerance of 0.05 min, and three minimum data points before removing duplicates with a $m/z$ tolerance of 1.5 ppm and retention time tolerance of 0.04 min. MetaCorrelate and ion identity networking were performed before exporting the final files. GNPS, SIRIUS, and feature information (legacy MZmine 2) modules were used to generate the feature table containing peak areas,.mgf files, and the content of each feature, respectively, which were necessary for downstream analysis.

## Data analysis

All tables generated using MZmine software were imported into R 4.4.1 for downstream analysis.

## Synthesis of Asn-UDCA

Solid ursodeoxycholic acid (1.27 mmol, 500 mg, 3 eq.) and 5 mL of DMF were added to a 20 mL glass vial with a stir bar. Next, solid EDC (1.27 mmol, 244 mg, 3 eq.) and neat DIPEA (4.25 mmol, 740 μL, 10 eq.) were subsequently added, and the solution was stirred at room temperature. After 15 min, asparagine (845 μmol, 112 mg, 2 eq.), DMAP were added, and the reaction was stirred overnight. The mixture was then concentrated in vacuo and purified by CombiFlash NextGen 300+

using reversed phase column C18 15.5 g Gold at a flow rate13 mL per min with $H_2O$ (Solvent A) and ACN (solvent B) using the following gradient: 0–6 min, 5% B; 6–10 min, 20% B; 10–17 min 20% B; 17–20 min, 30% B; 20–30 min, 30% B; 30–34 min, 40% B; 34–42 min, 40% B; 42–50 min, 80% B; 50–55 min, 80% B. Asn-UDCA eluted at 34 min, 40% B.

## LPS stimulation experiment

**Whole blood pre-treatment and stimulation ex vivo for phospho-Flow cytometry analysis.** A blood sample was obtained from healthy adult volunteers in BD Vacutainer Heparin Plasma Tubes. The blood sample was mixed in an equal ratio with WB-STIM buffer (Cytodelics AB) at room temperature. The sample was then split into groups: (i) non-treated, (ii) vehicle control treated with 0.25% dimethylsulfoxide (DMSO; Sigma-Aldrich), (iii) treated with each metabolite (50 μM conjugated UDCAs, collaborator, Finland), (iv) treated with kinase inhibitors Losmapimod (1500 nM, p38 inhibitor) and Selumetinib (1000 nM, ERK inhibitor), or (v) with combinations of both kinase, used as positive control. Metabolite-treated samples were incubated for 3 h, and kinase inhibitor-treated samples were incubated for 45 min at 37 °C and 5% $CO_2$. After incubation, samples were stimulated ex vivo with LPS (1 ng/mL) derived from Salmonella enterica for 5 time points (3, 7, 15, 30, and 60 min) at 37 °C and 5% $CO_2$. Whole blood samples were then fixed, red blood cells lysed, and samples washed using a Whole blood processing kit (Cytodelics AB)[36] according to the manufacturer's instructions. Completely fixed and processed white blood cells (WBCs) (0.5–1 × 10⁶ cells per sample) were plated and cryogenically preserved using CRYO#20 buffer (Cytodelics AB).

**Detecting intracellular phosphorylated protein using flow cytometry.** Cryopreserved cells were quickly thawed at 37 °C and then kept on ice while counted using Cellaca MX (Nexcelom). The 0.5–1 × 10⁶ cells were plated into each well in a 96-well U-bottom plate. Next, the cells were stained overnight at 4 °C with antibodies targeting intracellular antigens (Table 1), washed, and acquired using a Symphony A3 analyzer equipped with an HTS system.

## CD4⁺CD25⁻ T cell isolation and Th17/iTreg differentiation

Studies with primary human CD4⁺ T cells were approved by the Finnish Ethics Committee. Oral informed consent was obtained from all donors prior to the onset of the study. Primary human mononuclear cells were isolated from the umbilical cord blood of healthy neonates (Turku University Central Hospital, Turku, Finland) using Ficoll-Paque PLUS (Cytiva, Cat# 17144003) density gradient centrifugation. CD4⁺ T cells were further enriched using CD4⁺ Dynal positive selection beads (Invitrogen, Cat# 11331D) followed by CD25⁺ T cell depletion using the CD25 Microbeads II kit (Miltenyi Biotec, Cat# 130-092-983), according to the manufacturer's instructions. Prior to activation, naïve CD4⁺CD25⁻ T cells from different donors, which were highly positive for CD45RA (FITC anti-CD45RA, BD Biosciences, Cat# 555488, RRI-D:AB_395879) and negative for CD45RO (PE anti-CD45RO, BD Biosciences, Cat# 555493, RRID:AB_395884), were characterized by flow cytometry and were pooled.

Th17 cell differentiation was performed as described previously in refs. 37,38. In brief, CD4⁺CD25⁻ T cells were activated with plate-bound

anti-CD3 (3.75 μg/mL; Beckman Coulter, Cat# IM1304; RRID:AB_131612) and soluble anti-CD28 (1 μg/mL; Beckman Coulter, Cat# IM1376; RRID:AB_131624) in X-vivo 20 serum-free medium (Lonza), supplemented with L-glutamine (2 mM, Sigma-Aldrich) and antibiotics (50 U/mL penicillin plus 50 μg/mL streptomycin; Sigma-Aldrich). Th17 cells were cultured in the presence of IL6 (20 ng/mL; Roche, Cat# 7270-IL), IL1β (10 ng/mL; R&D Systems, Cat# 201-LB) and TGFβ (10 ng/mL; R&D Systems, Cat# 240-B), in the presence of neutralizing anti-IFNγ (1 μg/mL; R&D Systems, Cat# MAB285; RRID:AB_2123306) and anti-IL4 (1 μg/mL; R&D Systems, Cat# MAB204; RRID:AB_2126745) to block Th1 and Th2 differentiation, respectively.

iTreg cells were cultured as described earlier in refs. 38,39, with minor changes. Briefly, CD4$^+$CD25$^-$ T cells were activated with plate-bound anti-CD3 (2.5 μg/mL, Beckman Coulter, Cat# IM1304; RRID: AB_131612) and soluble anti-CD28 (0.5 μg/mL, Beckman Coulter, Cat# IM1376; RRID: AB_131624) in X-vivo 15 serum-free medium (Lonza), supplemented with L-glutamine (2 mM), penicillin (50 U), and streptomycin (50 μg/ml) (all from Biowest). Treg cell differentiation was induced in the presence of TGF-β (10 ng/mL; R&D Systems) and IL-2 (12 ng/mL; R&D Systems). The dilution for each antibody was 1/100.

To study the effect of MCBAs on human Th17 and iTreg cell differentiation, 100 μM of either Asn-UDCA, Ser-CDCA, Tyr-CDCA, or UDCA (in DMSO), and DMSO as a control, were added to the Th17 and iTreg cell culture media at day 0, and cultured for 72 h. After differentiation, secreted IL-17a levels were determined from Th17 cell-culture supernatants at 72 h using the human IL-17a DuoSet ELISA kit (R&D Systems, Cat# DY317-05, DY008). Expression of RORC and RORA in Th17 cells at 72 h of differentiation was assessed by quantitative real-time PCR, as described previously in ref. 37. For iTreg, intracellular staining of Foxp3 was performed using the eBioscience™ Foxp3/Transcription Factor Staining Buffer Set (Thermo Scientific, Cat# 00-5523-00), according to the manufacturer's protocol. Cells were stained with PE-conjugated Foxp3 antibody (Thermo Fisher Scientific, clone PCH101, Cat# 12-4776-42, RRID: _AB151878) or corresponding isotype control antibody (Thermo Fisher Scientific Cat# 12-4321-42, RRID: AB_1518773). Cells were incubated with fluorochrome-labeled antibody for 30 min at 4 °C. After staining, the cells were washed twice, resuspended in flow buffer (2% FBS/0.1% Na-azide/PBS), and acquired on BD LSRFortessa (BD Biosciences). The data was analyzed with FlowJo software (FlowJo LLC).

## Statistical methods

The metabolites data values were log-transformed prior to analysis. The difference in the lipidome and metabolome between the study groups was compared using a multivariate linear model. For longitudinal samples, linear mixed effects models were regressed with fixed effect (~sex + case + age + breastfeeding duration) and random effect ~ (1|Subject). For age-wise comparisons, the metabolites were regressed with various factors such as sex, and disease conditions (e.g., P1Ab vs CTR) using the MaAsLin2 package in R (BAs ~ sex + case). To subsequently visualize the metabolite level, forest plots from the ggplot2 R package were used. We used the paired t-test to analyze IL-17A and Foxp3 expression data, and the figures were plotted with GraphPad Prism8 software.

## Data availability

The targeted bile acid metabolomics datasets generated in this study are available in the MassIVE Repository (https://massive.ucsd.edu/ProteoSAFe/static/massive.jsp). These data can be accessed directly through GNPS/MassIVE under the accession number MSV000098739. MassIVE is a community resource developed by the NIH-funded Center for Computational Mass Spectrometry. Metagenomic sequencing data are available from the DIABIMMUNE project (https://diabimmune. broadinstitute.org/diabimmune/) under NCBI BioProject ID: PRJNA231909. Source data added to Figshare: https://doi.org/10.6084/m9.figshare.30408490.

## Code availability

R scripts and codes can be downloaded from: https://github.com/LamichhaneSantosh/MCBA_DIABIMMUNE. Any additional information required to reanalyze the data reported in this work paper is available from the lead contact upon request.

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

## Acknowledgements

This study was supported by the "Inflammation in human early life: targeting impacts on life-course health" (INITIALISE) consortium funded by the Horizon Europe Program of the European Union under Grant Agreement 101094099 (to M.O.), the Research Council of Finland funding (no. 363417 to S.L.; grants no. 331793 and 329277 to R.L.), the Research Council of Finland project grant (grant no. 333981 to M.O.), and the Novo Nordisk Foundation (grant no. NNF20OC0063971 to T.H. and M.O. and grant no. NNF19OC0057218 to R.L). Further support was received from the Swedish Research Council (grant no. 2016-05176 to T.H. and M.O.), the Sigrid Jusélius Foundation, the Jane and Aatos Erkko Foundation (to R.L.), Formas (grant no. 2019-00869 to T.H. and M.O.), and the Medical Research Funds, Helsinki University Hospital (to M.K.). Further support was received by the InFLAMES Flagship Programme of the Academy of Finland (grant no. 337530). We further acknowledge support by NICHD, P50HD106463, and NIDDK R01DK136117, U24DK133658 (to P.C.D.). RJX is funded by NIH (DK 43351, AI172147) and the Center for Microbiome Informatics and Therapeutics. Mass spectrometric analysis was performed at the Turku Metabolomics Centre with the support of Biocenter Finland. The DIABIMMUNE study was supported by the European Union Seventh Framework Programme FP7/2007-2013 under grant agreement number 202063 (to M.K. and R.L.), Breakthrough T1D (formerly Juvenile Diabetes Research Foundation) grants 17-2011-529 and 17-2014-305 to R.L., and the Academy of Finland Centre of Excellence in Molecular Systems Immunology and Physiology Research grant Decision number 250114, 2012–2017 (to M.O., R.L., and M.K.). We thank all voluntary blood donors and personnel of Turku University Hospital, Department of Obstetrics and Gynecology, Maternity Ward (Hospital District of Southwest Finland) for the umbilical cord blood collection. We acknowledge Marjo Hakkarainen (Turku Bioscience Center, University of Turku) and Sarita Heinonen for her excellent technical assistance. We thank Denys Mavrynsky and Tuomas Karskela for the synthesis of pure UDCA conjugates. We also acknowledge Mauricio Caraballo (Skaggs School of Pharmacy and Pharmaceutical Sciences, UC San Diego) and Nicole Avalon (Scripps Institution of Oceanography) for their assistance with the experimental setup for metabolomics and compound purification.

## Author contributions

Conceptualization: S.L. and M.O. Data curation: M.Kr., P.S., T.Lou, and P.K. Formal analysis: S.L., P.K., T.B, T.Lou, S.Z., and V.C.L. Funding acquisition: S.L., R.K., R.L., T.H., M.Kn, and M.O. Investigation: S.L., P.K., M.Kn., R.K., T.B., R.L. P.C.D., and M.O. Methodology: L.P., P.K., M.Kr., A.M.D., R.K., T.B., O.R., T.E.L., A.P., V.C.L., C.W., R.K., O.R., and T.H. Resources: T.V., R.J.X., A.P., P.C.D., E.C.G., P.B., A.A., J.M., A.M., T.L., M.R., K.Z., R.L., and M.Kn. Supervision: S.L., M.Kn., T.B., R.L., P.C.D., and M.O. Writing—original draft preparation: S.L. Writing—review and editing: all authors.

## Funding

## Competing interests

The author P.C.D. declares the following competing interests. P.C.D. is an advisor and holds equity in Cybele and Sirenas, is a science advisor and holds equity in bileOmix, and is a Scientific co-founder, advisor, and holds equity in Ometa, Enveda, and Arome, with prior approval by UC-San Diego. P.C.D. also consulted for DSM Animal Health in 2023. All other authors declare no competing interests.

## Additional information

[1]Turku Bioscience Centre, University of Turku and Åbo Akademi University, Turku, Finland. [2]Institute of Biomedicine, University of Turku, Turku, Finland. [3]InFLAMES Research Flagship Center, University of Turku, Turku, Finland. [4]Department of Chemistry, University of Turku, Turku, Finland. [5]Medical Research Council Laboratory of Medical Sciences (MRC LMS), Imperial College Hammersmith Campus, London, UK. [6]Department of Immunology and Inflammation, Imperial College London, London, UK. [7]Collaborative Mass Spectrometry Innovation Center, Skaggs School of Pharmacy and Pharmaceutical Sciences, University of California San Diego, San Diego, CA, USA. [8]Department of Pediatrics, University of California San Diego, La Jolla, CA, USA. [9]Virginia Tech, Department of Chemistry, VA, 24061 Blacksburg, USA. [10]Department of women's and children's health, Karolinska Institutet, Stockholm, Sweden. [11]Research Program for Clinical and Molecular Metabolism, Faculty of Medicine, University of Helsinki, Helsinki, Finland. [12]Liggins Institute, University of Auckland, Auckland, New Zealand. [13]Institute of Biotechnology, Helsinki Institute of Life Science, University of Helsinki, Helsinki, Finland. [14]The Broad Institute of MIT and Harvard, Cambridge, MA, USA. [15]Faculty of Agriculture and Forestry, University of Helsinki, Helsinki, Finland. [16]Center for Microbiome Innovation, University of California, San Diego, La Jolla, CA, USA. [17]Chiba University-UC San Diego Center for Mucosal Immunology, Allergy, and Vaccines (CU-UCSD cMAV), La Jolla, CA, USA. [18]Department of Bioengineering, University of California, San Diego, La Jolla, CA, USA. [19]School of Science and Technology, Örebro University, Örebro, Sweden. [20]Department of Pediatrics, Center for Child Health Research, Tampere University Hospital, Tampere, Finland. [21]Department of Life Technologies, University of Turku, Turku, Turku, Finland. [22]School of Medical Sciences, Faculty of Medicine and Health, Örebro University, Örebro, Sweden. ✉e-mail: santosh.lamichhane@utu.fi; mikael.knip@helsinki.fi; Matej.Oresic@oru.se

