## [Transparent Peer Review file · Nature Communications]

Microbiome derived bile acid signatures in early life and their association with islet autoimmunity

Corresponding Author: Dr Santosh Lamichhane

Version 0:

Reviewer comments:

Reviewer #1

(Remarks to the Author)

In this study, Lamichhane et al presented an investigation on early life microbiome derived bile acids on islet autoimmunity. While the study design is unique with repeated fecal sample collection up to six time points, there are many limitations downplaying our enthusiasm to support the publication in this present form.

Major comments:

- (1) Bile acids were measured only in fecal samples without a paired measurement in blood. While conjugated BAs are known as microbial related, their systemic bioavailability in blood matters, especially concerning their role in peripheral immune modulation. I would encourage the authors to show their data in blood to verify their main findings.
- (2) The longitudinal analysis includes only 74 children, though a total of 303 samples were provided. This sample size might be too small in terms of the multiple comparisons conducted in this work. However, the authors didn't correct any false discoveries in both human and experiment analyses.
- (3) The proposed trajectory analyses of microbial conjugated bile acids (MCBAs) are thoughtful. But the current analysis is limited to individual BAs rather than the overall BA profiles. The biological contributions of BAs to islet autoimmunity might be results of multiple BAs congeners rather than single BAs. I would suggest the authors exploring the trajectory patterns of all BAs and determining if any specific latent patterns exist. Meanwhile, maternal (e.g, diet, history of T1D or autoimmune diseases) and offspring feeding related factors (e.g., solid food introduction and other diets after breastfeeding stops) are not controlled in the analyses of MCBAs trajectory and differences across case status.
- (4) While the in vitro immune modulation experiments are compelling, the study does not mechanistically link MCBA alterations to in vivo immune dysfunction in progression to autoimmunity. For example: Are MCBA changes causally linked to autoantibody development, or are they correlative biomarkers? And How do MCBA levels interact with HLA risk alleles or other genetic factors? In addition, the overlap between MCBA-associated microbes and case status-linked taxa warrants deeper exploration but is currently only described via correlation.
- (5) It is unclear which time points with available gut microbiome data.
- (6) MCBAs showed sex-specific differences, but this observation is not explored further in experimental setting. Do the authors believe that the MCBAs examined do not contribute to the known sex disparities in autoimmune risk (e.g., female bias in T1D)?

Minor comments:

- (1) Inconsistent usage of "MCBAs" vs. "MBCAs" (e.g., Abstract vs. Methods). Standardize to "MCBAs" throughout.
- (2) Figure 4: Clarify whether error bars represent standard deviation or standard errors in forest plots. The overlaps of error bars and dots are quite confusing.
- (3) It is unclear how the missing values of MCBAs during six time points were handled in analyses. Meanwhile, the transformations of MCBAs/microbiota should be clarified.
- (4) Please add LOD/LOQ data for the MCBAs quantifications.
- (5) Please highlight how the batch effects were adjusted for the longitudinal sample processing and omics data profiling.

(Remarks on code availability)

Reviewer #2

(Remarks to the Author)

(Remarks on code availability)

None.

Reviewer #3

(Remarks to the Author)

The study investigated alterations in fecal microbially conjugated bile acids (MCBAs) in children who developed one or more islet autoantibodies, compared to age-matched controls who remained seronegative, over the period from 3 to 36 months of age. A total of 330 fecal samples were collected, and 110 MCBAs were profiled alongside gut microbiome composition analyses. Perturbations in the levels of ursodeoxycholic acid (UDCA) and deoxycholic acid (DCA) derivatives were observed. Their potential immunomodulatory roles were evaluated through in vitro assays using T cells polarized toward either Th17 or regulatory T cell (Treg) phenotypes.

General Comments

The study presents an intriguing dataset; however, the manuscript remains predominantly descriptive. Below are specific points of concern:

Introduction and Mechanistic Insight:

The introduction notes that MCBAs may function as ligands for nuclear and membrane receptors such as FXR, GPBAR1, and PXR. However, the manuscript does not provide experimental data to support receptor binding or activation by UDCA or DCA derivatives. This constitutes a critical gap in linking bile acid alterations to immune signaling.

Figure 6 – Incomplete Mechanistic Elucidation:

The data presented in Figure 6 appear preliminary. There is insufficient mechanistic explanation regarding how the tested bile acids influence intracellular signaling cascades—specifically the phosphorylation of p38, ERK, and degradation of I κ B α in LPS-stimulated monocytes.

Relevance to Th17 Biology – ROR γ t Modulation:

It is well established that several secondary bile acids can function as inverse agonists of ROR γ t, a transcription factor key to Th17 lineage commitment. The study should assess whether Asn-UDCA, Ser-UDCA, Tyr-CDCA, and UDCA modulate ROR γ t activity in T cells, which would substantiate claims related to Th17 polarization.

Foxp3 Modulation – Unexplored Mechanism:

The mechanisms by which the tested bile acids influence Foxp3 expression and Treg differentiation, are not explored in sufficient detail (Figure 7). Clarification of this mechanism is necessary to support the proposed functional conclusions.

Overinterpretation of Non-significant Findings (Figure 7):

Several panels in Figure 7 (specifically b, c, d, f, h, and i) do not reach statistical significance. These data should be interpreted with caution, and conclusions must reflect the actual statistical weight of the findings.

On the same line the in vitro modulation do not provide any insight on how these MCBA could modulate immunity in children.

(Remarks on code availability)

The link does not work

Reviewer #4

(Remarks to the Author)

The authors have used stool samples from an existing longitudinal observational, case-control study involving young children that did or did not develop one or more islet autoantibodies. They hypothesized that MCBA's formed by the gut microbiome can affect the occurrence of auto-immunity resulting in type 1 diabetes. They measured MCBA's in stool samples at 6 timepoints (3-36 months) and associated these to microbiome composition. To gain more mechanistic insights, they assed immune-modulatory properties of the MCBA's. They identified clear differences in stool MCBA's in children with single or multiple auto-antibodies and those with no antibodies and associated these differences to specific gut microbes. Additionally they identified immuno-modulatory properties of MBCA's on both innate and adaptive immune cells that might play a role in auto-immunity. The authors conclude that MCBAs influence immune development and type 1 diabetes risk. Overall this is a comprehensive study that indeed provides evidence that MCBAs produced by the gut microbiome have immunomodulatory affects and are possibly involved in developing type 1 diabetes.

Strengths include: the unique study cohort consisting of young children, the longitudinal study design and the state-of-the-art techniques used. Weaknesses include the rather low study power and potential selection issues.

A number of questions and comments:

1. Study subjects: 74 subjects were selected based on sample availability of infants at risk of type 1 diabetes. How were these 74 subjects selected, were these all available samples or a subset? Line 319 states that samples were collected longitudinally for each child but in figure 1 the largest number of samples for a timepoint is 55. This suggests several missing samples per timepoint, how was accounted for these missing samples? Specifically in the results presented in figure 4.
2. Results:
 - a. Are the data depicted in figure 4a an average of all timepoints?
 - b. I understand the authors choose to depict the unadjusted p-value in this figure as the groups are quite small. However, I would prefer to also include the adjusted p-value in the results section describing these data (line 143-153).
 - c. I suggest to include a statement on the relatively low power of this study in the discussion.
3. Results: Authors find a modulatory effect of MCBAs on LPS-induced signaling in human monocytes. These experiments were performed in whole blood samples, therefore including other cell types than monocytes. What was the reason for solely including the monocyte results? As these experiments are performed in whole blood, I assume that other immune cells are also measured. As T-cells are crucial in the development of type 1 diabetes it would be interesting to include information on these cells. Did MCBAs also modify LPS signaling in other immune cells such as T-cells?
4. Statistical analysis: The linear regression model was corrected for sex, age, and breast feeding duration. Where these factors also taken into account in the multivariate model (figure 4a and b)? Where these factors different between the 3 studie groups?
5. Statistical analysis: A spearman correlation analysis was performed to determine the link between bacterial composition and MCBA levels. No levels of significance are included in the results section or depicted in figure 5. I would suggest to include this to clarify the message
6. Statistical analysis: The statistical method section does not mention the method used for the analysis performed on IL17A and Foxp3 production/expression (figure 7), this could be included.
7. Line 155 to 164 are hard to read, what is meant by: "There was no persistent trend with respect to MCBA differences..."? Which is followed by a description of the differences in MCBAs at different timepoints. What do the authors mean by "no persistent trends"? Please clarify.
8. Line 165: MBCA should be MCBA, line 168: microbial conjugated BA profiles should be MCBA profiles, line 171: stain should be strain
9. Figure legends: Not in all figure legends subfigures (e.g. a,b,c) are specifically mentioned, please include this. Additionally the style is not consistent (for examples a. or (a)).

(Remarks on code availability)

Reviewer #5

(Remarks to the Author)

(Remarks on code availability)

Version 1:

Reviewer comments:

Reviewer #1

(Remarks to the Author)

The authors addressed most of the previous comments, but lack of blood bile acid measurements is still a major limitation.

(Remarks on code availability)

Reviewer #2

(Remarks to the Author)

(Remarks on code availability)

N.A

Reviewer #3

(Remarks to the Author)

The authors have responded to my questions/suggestions. I have no further comments.

(Remarks on code availability)

Reviewer #4

(Remarks to the Author)

Overall the authors have addressed the comments sufficiently. The only concern that remains is the reporting of solely unadjusted p-values in figure 4 and figure 5 and the main text, which can result in overinterpretation of the findings. I suggest to mention in the results section that findings are not significant after correction for multiple testing. Please also include adjusted p-values of the correlations shown in figure 5 in the supplement (as done for the data shown in figure 4).

(Remarks on code availability)

Reviewer #5

(Remarks to the Author)

(Remarks on code availability)

"I co-reviewed this manuscript with one of the reviewers who provided the listed reports. This is part of the Nature Communications initiative to facilitate training in peer review and to provide appropriate recognition for Early Career Researchers who co-review manuscripts."

Trajectories of microbiome-derived bile acids in early life – insights into the progression to islet autoimmunity" to Nature Communications. We have now received reports from 5 reviewers and, based on their comments, we have decided to invite a revision of your work. Your revision should address all the points raised by our reviewers (see their reports below). In particular, we would like to see in a revised version of the manuscript in which all technical points are addressed, with an improved statistical analysis, additional data regarding possible mechanism linking bile acid profiles to immune modulation suggested by #R1 and #R3, additional analyses on overall BA profiles and sex-specific differences requested by #R1. While we do not expect a full elucidation of causal links for how bile acid regulates immune signaling through nuclear and membrane receptor binding, we expect to see additional data on how MCBA alterations may modulate autoimmunity and T cell differentiation.

REVIEWER COMMENTS

Reviewer #1 (Remarks to the Author):

In this study, Lamichhane et al presented an investigation on early life microbiome derived bile acids on islet autoimmunity. While the study design is unique with repeated fecal sample collection up to six time points, there are many limitations downplaying our enthusiasm to support the publication in this present form.

Major comments:

(1) Bile acids were measured only in fecal samples without a paired measurement in blood. While conjugated BAs are known as microbial related, their systemic bioavailability in blood matters, especially concerning their role in peripheral immune modulation. I would encourage the authors to show their data in blood to verify their main findings.

Response: We agree with the reviewer that the systemic bioavailability of metabolite in blood is crucial. Due to lack of sample availability, we were not able to show these data in the circulating metabolite. However, we performed immune assay to understand how these bile acids modulate host immune responses. We acknowledge some limitations to our study, which we have updated in the revised manuscript (Page 15, Line 330-335).

(2) The longitudinal analysis includes only 74 children, though a total of 303 samples were provided. This sample size might be too small in terms of the multiple comparisons conducted in this work. However, the authors didn't correct any false discoveries in both human and experiment analyses.

Response: We did correct for false discoveries. The p values shown in figure are nominal p values, and the adjusted ones (for multiple comparisons using Benjamini-Hochberg) are shown in supplementary tables. We have clarified this information, in method section, figure legend and also added a discussion section in the revised manuscript relating to the sample size (Page 33, 751- 753 and Page 15, Line 330-335).

(3) The proposed trajectory analyses of microbial conjugated bile acids (MCBAs) are thoughtful. But the current analysis is limited to individual BAs rather than the overall BA profiles. The biological contributions of BAs to islet autoimmunity might be results of multiple BAs congeners rather than single BAs. I would suggest the authors exploring the trajectory patterns of all BAs and determining if any specific latent patterns exist. Meanwhile, maternal (e.g, diet, history of T1D or autoimmune diseases) and offspring feeding related factors (e.g., solid food introduction and other diets after breastfeeding stops) are not controlled in the analyses of MCBAs trajectory and differences across case status.

Response: We agree that diet is crucial for the bile acids, for that reason we conducted our analyses using in samples where breastfeeding information was available. Here, the breastfeeding duration was taken into account in our linear mixed model as suggested by the reviewer. We don't have data related to maternal diet.

(4) While the in vitro immune modulation experiments are compelling, the study does not mechanistically link MCBA alterations to in vivo immune dysfunction in progression to autoimmunity. For example: Are MCBA changes causally linked to autoantibody development, or are they correlative biomarkers? And How do MCBA levels interact with HLA risk alleles or other genetic factors? In addition, the overlap between MCBA-associated microbes and case status-linked taxa warrants deeper exploration but is currently only described via correlation.

Response: Our results demonstrate that the altered MCBAs modulate monocyte activation in response to immunostimulatory lipopolysaccharide and influence the Th17/Treg cell balance. Additionally, bacterial monoculture screening revealed that specific gut microbes are responsible for transforming these bile acids. Previous studies have suggested that such immunomodulatory effects may influence immune development and the risk of type 1 diabetes (T1D). While our findings offer potentially clinically significant insights into how microbial metabolites may affect T1D progression, the causal relationship remains unclear and warrants further investigation, which is beyond the scope of this study. We agree the reviewer and therefore, modified our discussion (Page 13-15).

(5) It is unclear which time points with available gut microbiome data.

Response: These were paired longitudinal sample (110) with the microbial conjugated BA profiles obtained at 6, 12, 18, 24, and 36 months of age from children. We have clarified this information in the revised manuscript (Page 8).

(6) MCBAs showed sex-specific differences, but this observation is not explored further in experimental setting. Do the authors believe that the MCBAs examined do not contribute to the known sex disparities in autoimmune risk (e.g., female bias in T1D)?

Response: The MCBAs examined may potentially contribute to the known sex disparities in autoimmune risk. To account for this, we included sex as a factor in our longitudinal model. Our analysis revealed that age was the most significant determinant of MCBA stool profiles, compared to sex (male vs. female) and case status (case vs. control). These effects are detailed in Supplementary Tables 1 (age), 2 (case status), and 3 (sex).

Minor comments:

(1) Inconsistent usage of "MCBAs" vs. "MBCAs" (e.g., Abstract vs. Methods). Standardize to "MCBAs" throughout.

Response: Thank you very much for pointing this out. This has been clarified in the revised version of the manuscript.

(2) Figure 4: Clarify whether error bars represent standard deviation or standard errors in forest plots. The overlaps of error bars and dots are quite confusing.

Response: This error bar is only plotted for the significant metabolites and this information has been clarified in the revised version of the manuscript.

(3) It is unclear how the missing values of MCBAs during six time points were handled in analyses. Meanwhile, the transformations of MCBAs/microbiota should be clarified.

Response: The MCBAs values were imputed using half the minimum detected value. For the Spearman correlation analysis, zero values in the microbiome data were retained as it is, given that Spearman is a rank-based method and not sensitive to the absolute values.

(4) Please add LOD/LOQ data for the MCBAs quantifications.

Response: MCBA identification was based on retention time matching using an in-house library generated from standards synthesized and provided by the Dorrestein Laboratory at UCSD. The concentrations of the standards were unknown, so we were unable to determine the limit of detection or limit of quantification for MCBA measurements.

(5) Please highlight how the batch effects were adjusted for the longitudinal sample processing and omics data profiling.

Response: No noticeable batch effects were observed in our data.

Reviewer #2 (Remarks to the Author):

Reviewer #2 (Remarks on code availability):

Response: The code is available at https://github.com/LamichhaneSantosh/MCBA_DIABIMMUNE

Reviewer #3 (Remarks to the Author):

The study investigated alterations in fecal microbially conjugated bile acids (MCBAs) in children who developed one or more islet autoantibodies, compared to age-matched controls who remained seronegative, over the period from 3 to 36 months of age. A total of 330 fecal samples were collected, and 110 MCBAs were profiled alongside gut microbiome composition analyses. Perturbations in the levels of ursodeoxycholic acid (UDCA) and deoxycholic acid (DCA) derivatives were observed. Their potential immunomodulatory roles were evaluated through in vitro assays using T cells polarized toward either Th17 or regulatory T cell (Treg) phenotypes.

General Comments

The study presents an intriguing dataset; however, the manuscript remains predominantly descriptive. Below are specific points of concern:

Introduction and Mechanistic Insight:

The introduction notes that MCBAs may function as ligands for nuclear and membrane receptors such as FXR, GPBAR₁, and PXR. However, the manuscript does not provide experimental data to support receptor binding or activation by UDCA or DCA derivatives. This constitutes a critical gap in linking bile acid alterations to immune signaling.

Response: We appreciate the reviewer's suggestion. While our introduction highlights the potential for MCBAs to act on nuclear and membrane receptors such as FXR, GPBAR₁, and PXR, we acknowledge that the current study does not directly assess receptor binding or activation. Our primary focus was to investigate the developmental trajectories of MCBAs during early life, including how these microbially derived bile acids are regulated in children who later progressed to islet autoimmunity. We aimed to identify candidate MCBAs within this context and offer an initial insight into their potential downstream effects on Th17 and Treg differentiation, thereby laying the groundwork for future mechanistic studies. We agree that defining specific receptor interactions is essential for fully elucidating the mechanisms of MCBA-mediated immune modulation. Future work will incorporate receptor activation

assays (e.g., reporter systems and binding affinity studies) to determine whether these derivatives directly engage FXR, GPBAR₁, PXR, or ROR α .

Figure 6 – Incomplete Mechanistic Elucidation:

The data presented in Figure 6 appear preliminary. There is insufficient mechanistic explanation regarding how the tested bile acids influence intracellular signaling cascades—specifically the phosphorylation of p38, ERK, and degradation of I κ B α in LPS-stimulated monocytes.

Response: We appreciate the reviewer's valuable feedback regarding the mechanistic elucidation in Figure 6. Our experiments demonstrated the immunomodulatory potential of various microbially conjugated and unconjugated bile acids on LPS-induced TLR₄ signaling in human monocytes within a physiologically relevant whole blood context, showing distinct effects on p38, ERK phosphorylation, and I κ B α degradation. We acknowledge that the precise molecular mechanisms by which these specific bile acids influence intracellular signaling cascades, including their direct receptors and detailed crosstalk with TLR₄, are not fully elucidated in the current dataset. We agree that a comprehensive mechanistic dissection is a critical next step. Future studies will focus on identifying specific receptors and elucidating the precise molecular interference points with the TLR₄ signaling pathway, which will be crucial for understanding this previously unrecognized layer of immune regulation in early life and T₁D development. We have revised the discussion to more explicitly state these limitations and future directions.

Relevance to Th₁₇ Biology – ROR γ t Modulation: It is well established that several secondary bile acids can function as inverse agonists of ROR γ t, a transcription factor key to Th₁₇ lineage commitment. The study should assess whether Asn-UDCA, Ser-UDCA, Tyr-CDCA, and UDCA modulate ROR γ t activity in T cells, which would substantiate claims related to Th₁₇ polarization.

Response: We thank the reviewer for this insightful suggestion. As noted, certain secondary bile acids act as inverse agonists of ROR γ t, the master regulator of Th₁₇ differentiation and IL-17A production (Hang et al., 2019). In line with the reviewer's recommendation, we assessed ROR γ t expression in T cells treated with Asn-UDCA, Ser-UDCA, Tyr-CDCA, and UDCA. However, neither TaqMan qPCR nor intracellular staining revealed significant changes in ROR γ t expression or correlation with IL-17A secretion.

While ROR γ t remains the canonical Th₁₇ transcription factor, ROR α (encoded by RORA) also contributes to IL-17A expression, particularly under inflammatory conditions in both mouse and human systems (Yang et al., 2008; Huh et al., 2011; Castro et al., 2017). Notably, we found that RORA expression positively correlated with IL-17A levels in Th₁₇ cells treated with the tested bile acids (Figure below), suggesting a potential role for ROR α in bile acid-mediated Th₁₇ modulation. Although ROR α has been studied in the context of circadian regulation and bile acid synthesis in mice (Ferrell et al., 2015), it is also a ligand-responsive nuclear receptor that may be directly modulated by bile acid derivatives, independently of circadian cues. This

raises the possibility that certain bile acids influence ROR α directly or via upstream metabolic signaling pathways, thereby promoting IL-17A expression. Our findings point to a potential ROR α -IL-17A axis in bile acid-driven immune regulation and highlight the need for future studies to define the upstream mechanisms that govern ROR α activation in this context. We have included RORA expression data in the revised manuscript (also shown below) and updated the corresponding figure legend and main text.

Figure 1: IL-17A secretion and RORA expression during Th17 cell differentiation treated with Asn-UDCA, Ser-UDCA, Tyr-CDCA, and UDCA. The upper panel shows IL-17A secretion, while the middle panel presents IL-17A levels normalized to DMSO control. The lower panel displays RORA gene expression in MCBA-treated Th17 cells measured by TaqMan qPCR at 72 hours of differentiation. Statistical significance was determined by paired t-test using four biological replicates.

Foxp3 Modulation – Unexplored Mechanism:

The mechanisms by which the tested bile acids influence Foxp3 expression and Treg differentiation, are not explored in sufficient detail (Figure 7). Clarification of this mechanism is necessary to support the proposed functional conclusions.

Response: We thank the reviewer for highlighting this important point. We would like to clarify that the tested bile acids exert a stronger and statistically significant effect on the Th17 phenotype, as indicated by IL-17A and RORA expression, whereas their effect on Foxp3 expression in Tregs showed an inverse but less pronounced trend. The mechanisms by which bile acids influence Foxp3 expression and Treg differentiation remain incompletely

understood, with only a few studies suggesting potential pathways. For instance, emerging evidence indicates that certain bile acid derivatives may enhance Treg function via mitochondrial reactive oxygen species (mtROS)-dependent chromatin remodeling at the *Foxp3* promoter, particularly through increased H3K27 acetylation (Hang et al., 2019; Kiriya & Nochi, 2023). Furthermore, the bile acid metabolite isoalloLCA has been shown to promote *Foxp3* expression through a mechanism involving chromatin remodeling that critically depends on the nuclear receptor NR4A1—highlighting a potential role for NR4A1 in bile acid-mediated Treg differentiation (10.1016/j.chom.2021.07.013). Although this lies beyond the scope of the present study, these mechanistic pathways, particularly those involving NR4A1 and epigenetic regulation, represent promising directions for future investigation into MCBA-driven immune modulation.

Overinterpretation of Non-significant Findings (Figure 7): Several panels in Figure 7 (specifically b, c, d, f, h, and i) do not reach statistical significance. These data should be interpreted with caution, and conclusions must reflect the actual statistical weight of the findings.

Response: We thank the reviewer for this helpful comment. We agree that statistical significance was not achieved in all panels referenced. Our intention was to present the measured expression and secretion levels of IL-17A and *Foxp3* as observed. To address inter-donor variability, values were normalized to the vehicle control (DMSO), which improved statistical resolution in several cases—most notably for IL-17A (Figure below, Panel A) and *RORA* expression (see above). *FOXP3* expression in Tregs treated with the respective MCBAs showed an inverse trend relative to IL-17A and *RORA* levels; however, these changes did not reach statistical significance (Figure below, Panel B). In response to the reviewer’s concern, we have included the normalized data in Supplementary Figure S4 and clarified this normalization approach in the corresponding figure legend. Additionally, we have revised the main text (Page 12, Line 255-260) to more cautiously interpret these findings, ensuring that our conclusions align with the statistical robustness of the data.

Figure 2: A) Levels of IL-17A secretion in Th17 cells treated with corresponding MCBAs (upper panel) and IL-17A levels normalized to DMSO control (lower panel). B) Mean fluorescence

intensity (MFI) of FOXP₃ expression in Tregs measured by flow cytometry (upper panel) and normalized to DMSO control (lower panel). Statistical significance was determined by paired t-test using four biological replicates.

On the same line the in vitro modulation does not provide any insight on how these MCBA could modulate immunity in children.

Response: We have modified discussion as per the context pointed by the reviewer.

Reviewer #3 (Remarks on code availability):

Response: The code is available at https://github.com/LamichhaneSantosh/MCBA_DIABIMMUNE

Reviewer #4 (Remarks to the Author):

The authors have used stool samples from an existing longitudinal observational, case-control study involving young children that did or did not develop one or more islet autoantibodies. They hypothesized that MCBA's formed by the gut microbiome can affect the occurrence of auto-immunity resulting in type 1 diabetes. They measured MCBA's in stool samples at 6 timepoints (3-36 months) and associated these to microbiome composition. To gain more mechanistic insights, they assed immune-modulatory properties of the MCBA's. They identified clear differences in stool MCBA's in children with single or multiple auto-antibodies and those with no antibodies and associated these differences to specific gut microbes. Additionally, they identified immuno-modulatory properties of MBCA's on both innate and adaptive immune cells that might play a role in auto-immunity. The authors conclude that MCBA's influence immune development and type 1 diabetes risk. Overall, this is a comprehensive study that indeed provides evidence that MCBA's produced by the gut microbiome have immunomodulatory affects and are possibly involved in developing type 1 diabetes. Strengths include: the unique study cohort consisting of young children, the longitudinal study design and the state-of-the-art techniques used.

Weaknesses include the rather low study power and potential selection issues.

A number of questions and comments:

1. Study subjects: 74 subjects were selected based on sample availability of infants at risk of type 1 diabetes. How were these 74 subjects selected, where these all-available samples or a subset? Line 319 states that samples were collected longitudinally for each child but in figure 1 the largest number of samples for a timepoint is 55. This suggests several missing samples per timepoint, how was accounted for these missing samples? Specifically in the results presented in figure 4.

Response: The DIABIMMUNE project was designed to test the hygiene hypothesis and investigate its role in the development of type 1 diabetes (T1D) and other autoimmune diseases in infants carrying HLA alleles that confer risk for islet autoimmunity and T1D. The subjects included in the present study were selected from a subset (n = 74) of available stool samples. As is common in many human longitudinal studies, it is logistically challenging to

obtain complete data points for all participants. To address this, we employed a linear mixed-effects model, incorporating age as a fixed effect and subject (with repeated measures) as a random effect, an established approach in the analysis of longitudinal data.

2. Results:

a. Are the data depicted in figure 4a an average of all timepoints?

Response: These are regression coefficient obtained from linear mixed effect models. We have clarified this information in the legend section of the revised manuscript.

b. I understand the authors choose to depict the unadjusted p-value in this figure as the groups are quite small. However, I would prefer to also include the adjusted p-value in the results section describing these data (line 143-153).

Response: This is relevant comment by the reviewer. The p-values shown are nominal; adjusted p-values (corrected for multiple comparisons using the Benjamini-Hochberg method) are available in Supplementary Table. We have clarified this information in the legend section of the revised manuscript.

c. I suggest to include a statement on the relatively low power of this study in the discussion.

Response: We have addressed this concern of the reviewer in the discussion section of the manuscript.

3. Results: Authors find a modulatory effect of MCBAs on LPS-induced signaling in human monocytes. These experiments were performed in whole blood samples, therefore including other cell types than monocytes. What was the reason for solely including the monocyte results? As these experiments are performed in whole blood, I assume that other immune cells are also measured. As T-cells are crucial in the development of type 1 diabetes it would be interesting to include information on these cells. Did MCBA's also modify LPS signaling in other immune cells such as T-cells?

Response: We appreciate the reviewer's valuable comment regarding the scope of our cellular analysis in whole blood samples. Our primary focus on CD14⁺ monocytes in this experiment was driven by their pivotal role as immediate and robust responders to LPS via TLR4, making them ideal for investigating rapid innate immune signaling events.

There are differential TLR4 responsiveness across immune cells. CD14⁺ monocytes express high levels of TLR4 and the co-receptor CD14, enabling a potent and rapid intracellular signaling cascade upon LPS exposure. This makes them central to the initial phases of TLR4 activation in whole blood, functioning as primary responders. While other innate immune cells like neutrophils also express TLR4, their responsiveness to LPS can differ significantly from monocytes. For instance, neutrophils express lower levels of CD14 compared to

monocytes, which can lead to a less potent signaling response to LPS, particularly at the early time points (0-60 minutes) assessed in our phospho flow assay. In contrast, T-cells are adaptive immune cells that do not express TLR4 as a primary LPS-sensing receptor. Their activation is indirect, requiring antigen presentation and co-stimulation from innate immune cells (like monocytes/APCs) over a much longer timeframe (hours to days). Consequently, our short-duration LPS stimulation (0-60 minutes) and phospho flow assay, designed for rapid innate immune signaling, are not suitable for directly evaluating T-cell responses to LPS. Any observed T-cell phosphorylation within this short window would likely be secondary to monocyte activation.

Our phospho flow cytometry pipeline was specifically optimized to investigate rapid, robust intracellular signaling in innate immune populations that directly respond to LPS in whole blood. Performing these experiments in whole blood provides a more physiologically relevant context than isolated PBMCs or cell lines. This allows us to study monocyte signaling in the presence of other circulating immune cells, plasma proteins, and other soluble factors that exist in the natural systemic environment, capturing complex intercellular interactions and soluble mediators that can influence immune responses.

4. Statistical analysis: The linear regression model was corrected for sex, age, and breast-feeding duration. Where these factors also taken into account in the multivariate model (figure 4a and b)? Where these factors different between the 3 studie groups?

Response: Yes, this has been taken into account in the multivariate model. Thank you very much for pointing this out.

5. Statistical analysis: A spearman correlation analysis was performed to determine the link between bacterial composition and MCBA levels. No levels of significance are included in the results section or depicted in figure 5. I would suggest to include this to clarify the message

Response: Taking reviewer comment into consideration, we have revised and updated the Figure 5 and shown the only those correlations coefficient with values with p less than 0.05.

6. Statistical analysis: The statistical method section does not mention the method used for the analysis performed on IL17A and Foxp3 production/expression (figure 7), this could be included.

Response: We thank the reviewer for pointing this out. We used the paired t-test to analyze IL-17A and Foxp3 expression data. This has now been clearly stated in the Statistical Analysis method section (Page 25).

7. Line 155 to 164 are hard to read, what is meant by: "There was no persistent trend with respect to MCBA differences..."? Which is followed by a description of the differences in

MCBAs at different timepoints. What do the authors mean by “no persistent trends”? Please clarify.

Response: This point has been clarified in the revised manuscript.

8. Line 165: MBCA should be MCBA, line 168: microbial conjugated BA profiles should be MCBA profiles, line 171: stain should be strain

Response: We have changed it as suggested by the reviewer.

9. Figure legends: Not in all figure legends subfigures (e.g. a,b,c) are specifically mentioned, please include this. Additionally, the style is not consistent (for examples a. or (a)).

Response: We have addressed this concern of the reviewer in the revised version of the manuscript.

Reviewer #5 (Remarks to the Author):

REVIEWERS' COMMENTS

Reviewer #1 (Remarks to the Author):

The authors addressed most of the previous comments, but lack of blood bile acid measurements is still a major limitation.

Response: We have acknowledged the lack of blood data as a limitation in the Discussion section of the manuscript.

Reviewer #2 (Remarks to the Author):

Reviewer #2 (Remarks on code availability):

N.A

Reviewer #3 (Remarks to the Author):

The authors have responded to my questions/suggestions. I have no further comments.

Reviewer #4 (Remarks to the Author):

Overall the authors have addressed the comments sufficiently. The only concern that remains is the reporting of solely unadjusted p-values in figure 4 and figure 5 and the main text, which can result in overinterpretation of the findings.

I suggest to mention in the results section that findings are not significant after correction for multiple testing. Please also include adjusted p-values of the correlations shown in figure 5 in the supplement (as done for the data shown in figure 4).

Response: We have added adjusted p values in the Supplementary tables. Also, we have acknowledged in our manuscript about the sample size. We have also clarified in the legend relating to the p and adjusted values.

Reviewer #5 (Remarks to the Author):

Reviewer #5 (Remarks on code availability):

"I co-reviewed this manuscript with one of the reviewers who provided the listed reports. This is part of the Nature Communications initiative to facilitate training in peer review and to provide appropriate recognition for Early Career Researchers who co-review manuscripts."